# A universal vector concept for a direct genotyping of transgenic organisms and a systematic creation of homozygous lines

**Frederic Strobl\*, Anita Anderl, Ernst HK Stelzer\***

Physical Biology, BMLS, CEF-MC, Goethe Universität, Frankfurt am Main, Germany

**Abstract** Diploid transgenic organisms are either hemi- or homozygous. Genetic assays are, therefore, required to identify the genotype. Our AGameOfClones vector concept uses two clearly distinguishable transformation markers embedded in interweaved, but incompatible Lox site pairs. Cre-mediated recombination leads to hemizygous individuals that carry only one marker. In the following generation, heterozygous descendants are identified by the presence of both markers and produce homozygous progeny that are selected by the lack of one marker. We prove our concept in *Tribolium castaneum* by systematically creating multiple functional homozygous transgenic lines suitable for long-term fluorescence live imaging. Our approach saves resources and simplifies transgenic organism handling. Since the concept relies on the universal Cre-Lox system, it is expected to work in all diploid model organisms, for example, insects, zebrafish, rodents and plants. With appropriate adaptions, it can be used in knock-out assays to preselect homozygous individuals and thus minimize the number of wasted animals.
DOI: https://doi.org/10.7554/eLife.31677.001

**\*For correspondence:**
frederic.strobl@
physikalischebiologie.de (FS);
ernst.stelzer@
physikalischebiologie.de (EHKS)

## Introduction

Life sciences, especially cell and developmental biology, rely on model organisms. The most frequently used vertebrates are mouse and zebrafish. Amongst insects, the fruit fly *Drosophila melanogaster* and the red flour beetle *Tribolium castaneum* are the two prevailing species. An important standard technique is transgenesis, that is, the insertion of recombinant DNA into the genome of the model organism (*Gama Sosa et al., 2010*). Since model organisms are typically diploid, the genotype has to be considered, which leads to a certain experimental complexity. Usual mating schemes result in (i) non-transgenic wild-type progeny, (ii) hemizygous transgenic progeny, that is, only the maternal or only the paternal chromosome carries the transgene, and (iii) homozygous transgenic progeny, that is, both the maternal and paternal chromosomes carry the transgene. In rare cases, the phenotype reveals the genotype, but usually, either two of the three or even all three outcomes cannot be distinguished. Transformation markers can be used to separate wild-type from transgenic, but not hemi- from homozygous individuals. Thus, additional experiments are necessary to determine the genotype, for example genetic assays, which are invasive and require manpower as well as consumables.

In our AGameOfClones (AGOC) vector concept, all genotypes are directly identifiable by specifically designed distinct phenotypes, which permits the systematic creation of homozygous transgenic lines. Our approach relies on two clearly distinguishable transformation markers embedded in interweaved, but incompatible Lox site pairs. Cre-mediated recombination results in hemizygous individuals that retain only one of the two markers and are thus phenotypically distinguishable from each other and the wild type. In the next generation, descendants that express both markers are identified as heterozygous for the transgene. Finally, a cross of two heterozygotes results in homozygous progeny that are selected by the lack of one marker.

**eLife digest** Researchers frequently use model organisms, such as mice, zebrafish and various insect species, to understand biological processes – with the underlying idea that discoveries made can be applied to other species too. A common technique is genetic manipulation, in which a foreign gene is inserted into the chromosome of an organism. These introduced genes are called transgenes and the organisms carrying them are referred to as transgenic. Transgenic organisms are powerful tools to analyze biological processes or mimic human diseases.

Many model organisms carry two homologous chromosomes – one inherited from each parent. Pairs of chromosomes carry genes in the same order, but do not necessarily have identical versions of those genes. Newly created transgenic organisms, however, carry the transgene on only one of the chromosomes. This can be a problem for researchers, as many experiments require individuals that carry the transgene on both. Unfortunately, only costly and error-prone methods can distinguish between these individuals.

To overcome these drawbacks, Strobl et al. developed a concept called AGameOfClones and applied it to the red flour beetle *Tribolium castaneum*. In their approach, the transgene also expresses two marker-proteins with different fluorescent colors. After several generations of breeding, two versions of the transgene emerge – each retaining only one of the markers. This means that in the following generation, descendants that express both markers must be the offspring that carry the transgene on both of the chromosomes.

The AGameOfClones concept has several major advantages: individuals with different markers can be easily identified, the procedure is cost-efficient and reliable, and it can be applied to nearly all model organisms. This will benefit breeding schemes and animal welfare since irrelevant individuals can be excluded as soon as the markers become detectable.
DOI: https://doi.org/10.7554/eLife.31677.002

## Results

### Proof-of-principle in the emerging insect model organism *Tribolium castaneum*

The proof-of-principle of the AGOC vector concept relied on the red flour beetle *Tribolium castaneum*, an emerging insect model organism (*Klingler, 2004*; *Brown et al., 2009*), in conjunction with the piggyBac transposon system (*Lorenzen et al., 2003*; *Berghammer et al., 2009*), which allows semi-random genomic insertion. We developed the transformation-ready pAGOC vector (*Figure 1—figure supplement 1*) that contains mOrange-based (*Shaner et al., 2008*) and mCherry-based (*Shaner et al., 2004*) eye-specific (*Berghammer et al., 1999*) transformation markers (mO and mC, respectively). Both fluorescent proteins are spectrally separable by appropriate excitation bands and emission filters (*Shaner et al., 2005*). Each marker is flanked upstream by a LoxP site (*Hamilton and Abremski, 1984*) and downstream by a LoxN (*Livet et al., 2007*) site, resulting in interweaved Lox site pairs (*Figure 1*). Due to variations in the spacer sequences, LoxP and LoxN sites are incompatible with each other.

We injected this vector together with a piggyBac transposase-expressing helper vector (that is, pATub'piggyBac) into pre-blastoderm embryos to achieve germline transformation. All survivors, that is, F1 potential mosaics, were mated with wild types and in six of these crosses, at least one F2 (mO-mC) founder female was found among the progeny. For each cross, one founder female was mated with a wild-type male and the progeny were scored to confirm that only a single insertion had occurred (*Supplementary file 1*). Transgenic descendants were collected to establish six proof-of-principle cultures, which carry the same transgene, but in different genomic locations. These F3 (mO-mC) pre-recombination hemizygous sublines were called AGOC #1 to #6. To roughly estimate homozygous viability, two F3 (mO-mC) pre-recombination hemizygous siblings were mated and the progeny were scored (*Supplementary file 2*). Additionally, the insertion locations of the transgenes were determined in four of the six AGOC sublines (*Supplementary file 3*). Up to this step, our scheme did not differ from most standard procedures to establish transgenic lines.

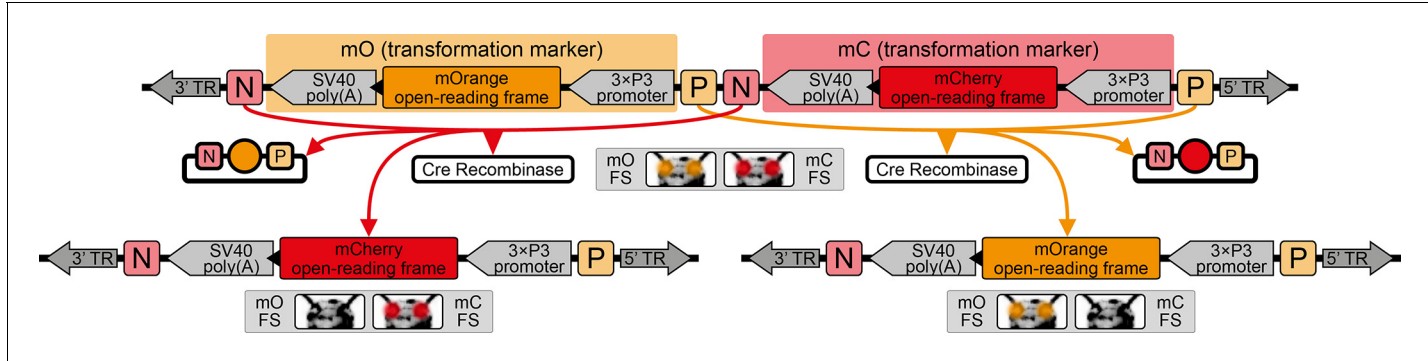

**Figure 1.** The AGameOfClones vector concept within the piggyBac-based transformation-ready pAGOC vector for *Tribolium*. Two fluorescence-based transformation markers, mO and mC, are embedded into a piggyBac-based transformation-ready vector, which is characterized by 3' and 5' terminal repeats (TR) necessary for genomic insertion. The markers are based on the artificial eye-specific 3×P3 promoter, the open-reading frame for the respective fluorescent protein, that is, mOrange or mCherry, and the SV40 poly(A). Each transformation marker is flanked upstream by a LoxP site (P) and downstream by a LoxN site (N), forming interweaved Lox site pairs. The markers can be detected in the eyes by using appropriate filter sets (FS). Cre-mediated recombination leads to the excision of one marker from the genome. Upon removal, the other marker remains within the genome, since the remaining LoxP and LoxN sites are incompatible. Individuals that underwent recombination give rise to progeny in which only one marker is detected in the eyes.

DOI: https://doi.org/10.7554/eLife.31677.003

The following figure supplements are available for figure 1:

**Figure supplement 1.** The pAGOC vector.
DOI: https://doi.org/10.7554/eLife.31677.004

**Figure supplement 2.** The pAGOC{#P'#O(LA)-mEmerald} vector.
DOI: https://doi.org/10.7554/eLife.31677.005

**Figure supplement 3.** The pAVOIAF{#1–#2–#3–#4} vector.
DOI: https://doi.org/10.7554/eLife.31677.006

**Figure supplement 4.** Development of the 24 vectors used in this study.
DOI: https://doi.org/10.7554/eLife.31677.007

## Systematic creation of homozygous transgenic lines

The mating procedure for the systematic creation of homozygous transgenic lines (*Figure 2*, an comprehensive scheme is provided in *Figure 2—figure supplement 1*) spanned four generations and involved a transgenic helper line, ICE{HSP68'NLS-Cre} #1. This line expresses a nuclear-localized Cre recombinase (*Peitz et al., 2002*) under control of the *heat shock protein 68b* promoter (*Schinko et al., 2012*) and carries a mCerulean-based (*Markwardt et al., 2011*) eye-specific transformation marker (mCe). The procedure was performed with all six AGOC sublines and phenotypically documented for #5 and #6 (*Figure 3*).

1. F3 (mO-mC) pre-recombination hemizygous females, which carried mO and mC on the maternal chromosome in cis configuration, were mated with (mCe/mCe) homozygous helper males (*Figure 2* and *Figure 3*, first row). This resulted in F4 (mCe; mO-mC) double hemizygotes in which Cre-mediated recombination occurs (*Table 1*, F3 row). In this hybrid generation, adults displayed a patchy expression of mO and mC within their compound eyes (*Figure 3—figure supplement 1*).
2. F4 (mCe; mO-mC) double hemizygous females were mated with wild-type males (*Figure 2* and *Figure 3*, second row). Due to Cre-mediated recombination in the germline, this resulted in F5 (mO) and (mC) post-recombination hemizygotes that carried either only mO or only mC on the maternal chromosome (*Table 1*, F4 row).
3. F5 (mO) post-recombination hemizygous females were mated with F5 (mC) post-recombination hemizygous male siblings (*Figure 2* and *Figure 3*, third row), which resulted in F6 (mO/mC) heterozygotes that carried mO on the maternal and mC on the paternal chromosome in trans configuration (*Table 1*, F5 row). This was demonstrated by mating F6 (mO/mC) heterozygous females with wild-type males and scoring the progeny (*Table 1*, F6-S row).

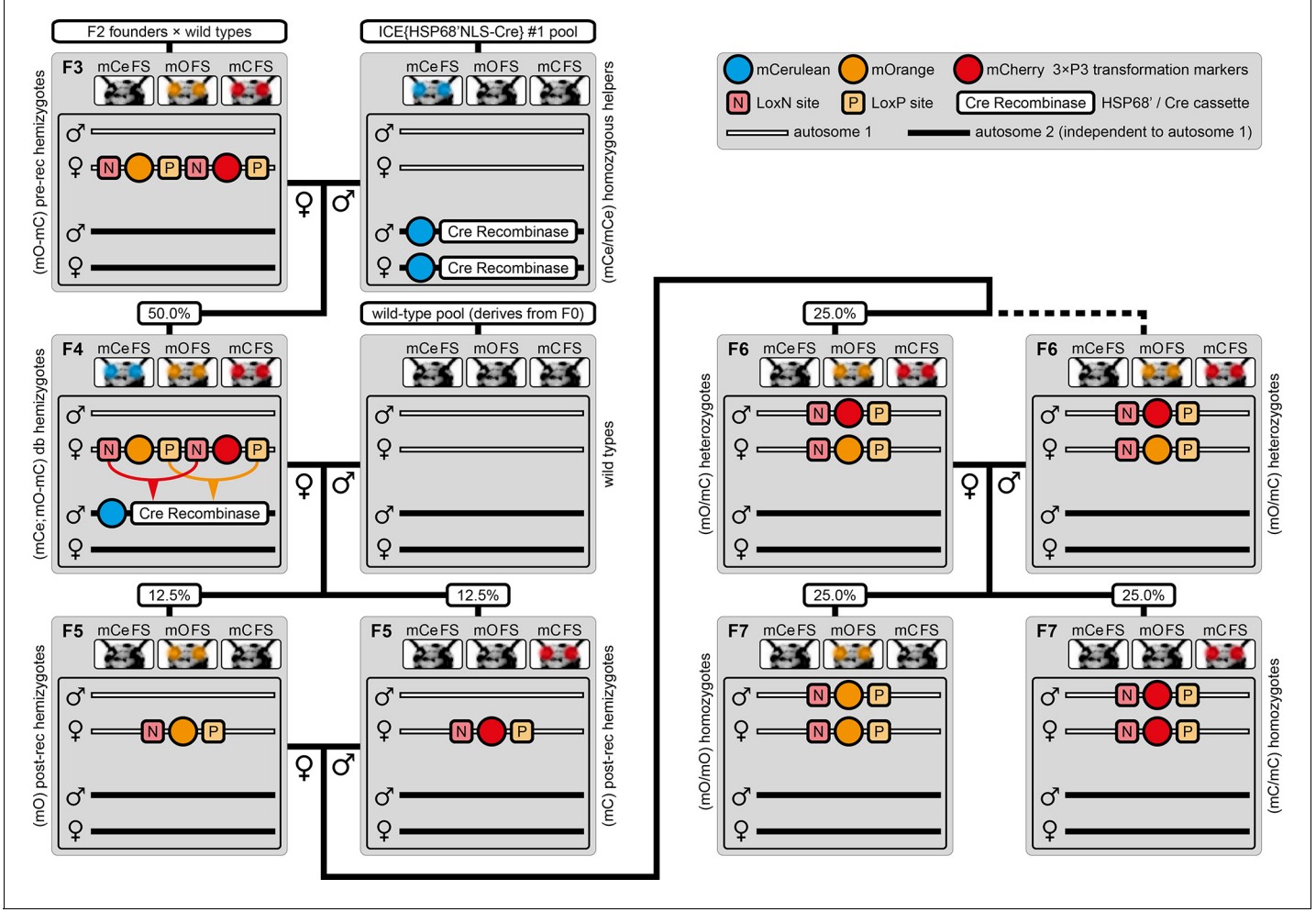

**Figure 2.** The AGameOfClones F3 to F7 mating procedure for the systematic creation of homozygous transgenic *Tribolium* lines. A rounded rectangle illustrates the genotype for two independent autosomes, white bars represent the AGOC transgene location and black bars the Cre recombinase-expressing helper transgene location. A F2 (mO-mC) founder female × wild-type male cross gives rise to F3 (mO-mC) pre-recombination hemizygotes that carry mO and mC in cis configuration. A F3 (mO-mC) pre-recombination hemizygous female × mCe homozygous helper male cross results in F4 (mCe; mO-mC) double hemizygotes, in which one marker is removed through Cre-mediated recombination. Next, a F4 (mCe; mO-mC) double hemizygous female × wild-type male cross gives rise to F5 (mO) and (mC) post-recombination hemizygotes. A F5 (mO) post-recombination hemizygous female × F5 (mC) post-recombination hemizygous male sibling cross results in F6 (mO/mC) heterozygous progeny that carry mO and mC in trans configuration. Finally, a F6 (mO/mC) heterozygous female × a F6 (mO/mC) heterozygous male sibling cross gives rise to F7 (mO/mO) and (mC/mC) homozygous progeny. The percentage boxes indicate the theoretical ratio of the progeny that carry the respective genotype, the dashed line represents genotypically identical siblings. FS, filter set; rec, recombination; db, double.

DOI: https://doi.org/10.7554/eLife.31677.008

The following figure supplements are available for figure 2:

**Figure supplement 1.** The AGameOfClones F3 to F7 mating procedure with all outcomes and respective Punnett squares.

DOI: https://doi.org/10.7554/eLife.31677.009

**Figure supplement 2.** Alternative AGameOfClones F3 to F7 mating procedure for transgenes located on the X allosome.

DOI: https://doi.org/10.7554/eLife.31677.010

4. F6 (mO/mC) heterozygous females were mated with genotypically identical male siblings (*Figure 2* and *Figure 3*, fourth row), which resulted in F7 (mO/mO) and (mC/mC) homozygotes that carried either only mO or only mC on both, the maternal and paternal chromosome (*Figure 2* and *Figure 3*, fifth row as well as *Table 1*, F6 row). This was demonstrated by mating F7 (mO/mO) and (mC/mC) homozygous females with wild-type males and scoring the progeny (*Table 1*, F7-O and F7-C row, respectively).

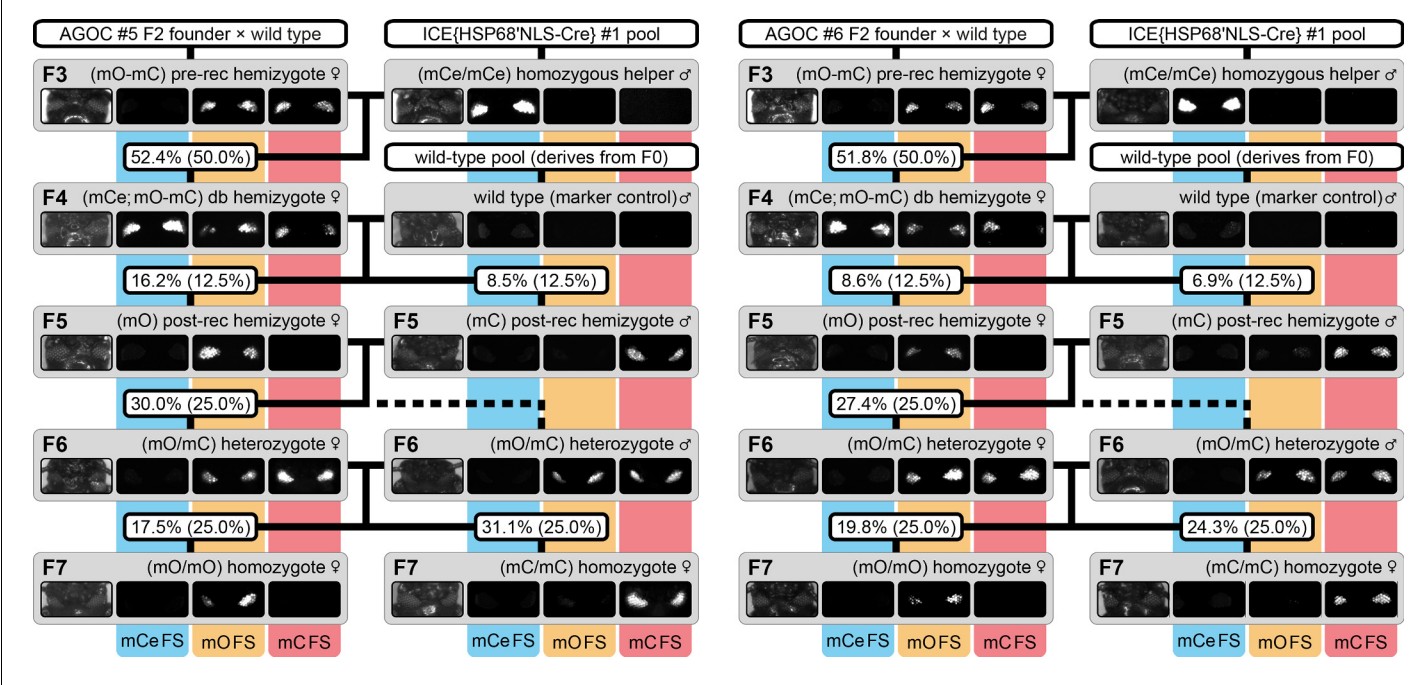

**Figure 3.** The AGameOfClones F3 to F7 mating procedure demonstrated for the AGOC #5 and #6 sublines. From the F3 to the F7 generation, the genotype was phenotypically determined by monitoring mCe, mO and mC. For both sublines, F7 (mO/mO) and (mC/mC) homozygotes were obtained by following the mating procedure outlined in *Figure 2*. The wild-type male in the second row functions as the marker control. The percentage boxes indicate the experimental (and theoretical) ratio of the progeny that displayed the respective phenotype. FS, filter set; rec, recombination; db, double.
DOI: https://doi.org/10.7554/eLife.31677.011

The following figure supplements are available for figure 3:

**Figure supplement 1.** The F4 (mCe; mO-mC) double hemizygous generation.

DOI: https://doi.org/10.7554/eLife.31677.012

**Figure supplement 2.** The AGameOfClones F3 to F7 mating procedure demonstrated for the AGOC #5 and #6 sublines with swapped genders as well as with an alternative Cre recombinase-expressing helper subline.

DOI: https://doi.org/10.7554/eLife.31677.013

Throughout all generations, the subline-specific scores matched the expectations. No significant differences between the respective arithmetic means and the theoretical Mendelian ratios were found. Importantly, all expected phenotypes, and thus all expected genotypes, were found in all generations and consequently, F7 (mO/mO) as well as (mC/mC) homozygotes were obtained for all six AGOC sublines. A description of the generations and their characteristics is found in *Supplementary file 4*.

Two controls were performed with the AGOC #5 and #6 sublines to confirm proper function of the AGOC vector concept: The F3 to F7 crossing procedure was successfully conducted with (i) swapped genders (*Figure 3—figure supplement 2A*, *Supplementary file 5*) and (ii) an alternative helper subline (*Figure 3—figure supplement 2B*, *Supplementary file 5*), ICE{HSP68'NLS-Cre} #2, which carries the same Cre recombinase-expressing cassette as the #1 subline, but at a different genomic location.

## Systematic creation of functional homozygous AGOC lines and sublines

Based on pAGOC, we developed the intermediate pAGOC{#P'#O(LA)-mEmerald} vector (*Figure 1—figure supplement 2*), which contains an open-reading frame for mEmerald-labeled (*Shaner et al., 2005*) Lifeact, a small and universal peptide tag derived from *Saccharomyces cerevisiae* that binds to filamentous actin (*Riedl et al., 2008*). With this vector, three transformation-ready derivates were created that allow expression of mEmerald-labeled Lifeact under control of either the *tubulin alpha*

**Table 1.** Mating procedure results for the six proof-of-principle AGOC sublines from the F3 to the F7 generation.

Bold entries mark progeny that were used in the subsequent cross. F6-S, F7-O and F7-C are control crosses. No significant differences between the arithmetic means and the theoretical Mendelian ratios were found. See Source Data 1 for raw scores ordered by transgenic sublines.

| Gen | Cross | Subline | ●●● | 🔵●● | ●🟠● | ●●🔴 | 🔵🟠● | 🔵●🔴 | ●🟠🔴 | 🔵🟠🔴 | Total |
|---|---|---|---|---|---|---|---|---|---|---|---|
| F3 | | Theoretical | - | 50.0% | - | - | - | - | - | **50.0%** | - |
| | | AGOC #1 | - | 47.9% (46) | - | - | - | - | - | **52.1% (50)** | 96 |
| | | AGOC #2 | - | 52.4% (44) | - | - | - | - | - | **47.6% (40)** | 84 |
| | | AGOC #3 | - | 55.1% (49) | - | - | - | - | - | **44.9% (40)** | 89 |
| | | AGOC #4 | - | 44.1% (41) | - | - | - | - | - | **55.9% (52)** | 93 |
| | | AGOC #5 | - | 47.6% (39) | - | - | - | - | - | **52.4% (43)** | 82 |
| | | AGOC #6 | - | 48.2% (27) | - | - | - | - | - | **51.8% (29)** | 56 |
| | | Mean | - | 49.2 ± 3.9% | - | - | - | - | - | **50.8 ± 3.9%** | 83.3 |
| F4 | | Theoretical | 25.0% | 25.0% | **12.5%** | **12.5%** | 12.5% | 12.5% | - | - | - |
| | | AGOC #1 | 25.6% (30) | 34.2% (40) | **4.3% (5)** | **10.2% (12)** | 6.0% (7) | 19.7% (23) | - | - | 117 |
| | | AGOC #2 | 27.3% (36) | 28.9% (38) | **15.1% (20)** | **8.3% (11)** | 9.8% (13) | 10.6% (14) | - | - | 132 |
| | | AGOC #3 | 25.8% (33) | 32.8% (42) | **13.3% (17)** | **10.1% (13)** | 5.5% (7) | 12.5% (16) | - | - | 128 |
| | | AGOC #4* | 14.0% (14) | 15.0% (15) | **20.0% (20)** | **13.0% (13)** | 22.0% (22) | 9.0% (9) | 3.0% (3) | 4.0% (4) | 100 |
| | | AGOC #5 | 17.1% (20) | 40.2% (47) | **16.2% (19)** | **8.5% (10)** | 10.3% (12) | 7.7% (9) | - | - | 117 |
| | | AGOC #6 | 33.6% (39) | 33.6% (39) | **8.6% (10)** | **6.9% (8)** | 6.1% (7) | 11.2% (13) | - | - | 116 |
| | | Mean | 23.9 ± 7.2% | 30.8 ± 8.5% | **12.9 ± 5.6%** | **9.5 ± 2.1%** | 9.9 ± 6.3% | 11.8 ± 4.2% | 0.5% | 0.7% | 118.3 |
| F5 | | Theoretical | 25.0% | - | 25.0% | 25.0% | - | - | **25.0%** | - | - |
| | | AGOC #1 | 27.2% (31) | - | 26.3% (30) | 23.7% (27) | - | - | **22.8% (26)** | - | 114 |
| | | AGOC #2 | 28.9% (26) | - | 33.3% (30) | 17.8% (16) | - | - | **20.0% (18)** | - | 90 |
| | | AGOC #3 | 24.8% (30) | - | 27.3% (33) | 24.8% (30) | - | - | **23.1% (28)** | - | 121 |
| | | AGOC #4 | 19.3% (21) | - | 22.9% (25) | 37.6% (41) | - | - | **20.2% (22)** | - | 109 |
| | | AGOC #5 | 28.2% (31) | - | 29.1% (32) | 12.7% (14) | - | - | **30.0% (33)** | - | 110 |
| | | AGOC #6 | 26.2% (22) | - | 29.7% (25) | 16.7% (14) | - | - | **27.4% (23)** | - | 84 |
| | | Mean | 25.8 ± 3.5% | - | 28.1 ± 3.5% | 22.2 ± 8.8% | - | - | **23.9 ± 4.0%** | - | 104.7 |
| F6-S | | Theoretical | - | - | 50.0% | 50.0% | - | - | - | - | - |
| | | AGOC #1 | - | - | 46.4% (39) | 53.6% (44) | - | - | - | - | 84 |
| | | AGOC #2 | - | - | 50.0% (49) | 50.0% (49) | - | - | - | - | 98 |
| | | AGOC #3 | - | - | 54.0% (68) | 46.0% (58) | - | - | - | - | 126 |
| | | AGOC #4 | - | - | 53.8% (50) | 46.2% (43) | - | - | - | - | 93 |
| | | AGOC #5 | - | - | 51.3% (59) | 48.7% (56) | - | - | - | - | 115 |
| | | AGOC #6 | - | - | 57.0% (49) | 43.0% (37) | - | - | - | - | 86 |
| | | Mean | - | - | 52.1 ± 3.7% | 47.9 ± 3.7% | - | - | - | - | 100.3 |
| F6 | | Theoretical | - | - | **25.0%** | **25.0%** | - | - | 50.0% | - | - |
| | | AGOC #1 | - | - | **20.3% (23)** | **24.8% (28)** | - | - | 54.9% (62) | - | 113 |
| | | AGOC #2 | - | - | **21.5% (23)** | **35.5% (38)** | - | - | 43.0% (46) | - | 117 |
| | | AGOC #3 | - | - | **22.9% (27)** | **22.9% (27)** | - | - | 54.2% (64) | - | 118 |
| | | AGOC #4 | - | - | **22.0% (29)** | **22.7% (30)** | - | - | 55.3% (73) | - | 132 |
| | | AGOC #5 | - | - | **17.5% (18)** | **31.1% (32)** | - | - | 51.4% (53) | - | 103 |
| | | AGOC #6 | - | - | **19.8% (22)** | **24.3% (27)** | - | - | 55.9% (62) | - | 111 |
| | | Mean | - | - | **20.7 ± 1.9%** | **26.9 ± 5.2%** | - | - | 52.4 ± 4.9% | - | 115.7 |

*Table 1 continued on next page*

*Table 1 continued*

| Gen | Cross | Subline | Progeny ●●● | ●●● | ●●● | ●●● | ●●● | ●●● | ●●● | ●●● | Total |
|-----|-------|---------|-----|-----|-----|-----|-----|-----|-----|-----|-------|
| F7-O | | Theoretical | - | - | 100% | - | - | - | - | - | - |
| | | AGOC #1 | - | - | 100% (94) | - | - | - | - | - | 94 |
| | | AGOC #2 | - | - | 100% (50) | - | - | - | - | - | 50 |
| | | AGOC #3 | - | - | 100% (49) | - | - | - | - | - | 49 |
| | | AGOC #4 | - | - | 100% (79) | - | - | - | - | - | 79 |
| | | AGOC #5 | - | - | 100% (79) | - | - | - | - | - | 79 |
| | | AGOC #6 | - | - | 100% (63) | - | - | - | - | - | 63 |
| | | Mean | - | - | 100 ± 0% | - | - | - | - | - | 69.0 |
| F7-C | | Theoretical | - | - | - | 100% | - | - | - | - | - |
| | | AGOC #1 | - | - | - | 100% (101) | - | - | - | - | 101 |
| | | AGOC #2 | - | - | - | 100% (105) | - | - | - | - | 105 |
| | | AGOC #3 | - | - | - | 100% (89) | - | - | - | - | 89 |
| | | AGOC #4 | - | - | - | 100% (54) | - | - | - | - | 54 |
| | | AGOC #5 | - | - | - | 100% (74) | - | - | - | - | 74 |
| | | AGOC #6 | - | - | - | 100% (64) | - | - | - | - | 64 |
| | | Mean | - | - | - | 100 ± 0% | - | - | - | - | 81.2 |

*In the AGOC #4 subline, incomplete recombination occurred in the F4 (mCe; mO-mC) double hemizygous generation, as we obtained several F5 individuals that still carried both transformation markers (7.0% in total). We continued the mating procedure with the F5 (mO) and (mC) post-recombination hemizygous progeny.

DOI: https://doi.org/10.7554/eLife.31677.014

The following source data available for Table 1:

**Source data 1.** Raw scores for all mating procedure result tables ordered by transgenic sublines.
DOI: https://doi.org/10.7554/eLife.31677.015

1-like protein (*Siebert et al., 2008*), the *zerknüllt 1* (*van der Zee et al., 2005*; ; *Sharma et al., 2013*; *Panfilio et al., 2013*; *Hilbrant et al., 2016*) or the *actin-related protein 5* promoter. Additionally, we developed two more derivates that contain expression cassettes for either mEmerald-labeled *beta-galactoside alpha-2,6-sialyltransferase 1* or *histone H2B* under control of the *tubulin alpha 1-like protein* promoter. With these vectors, we created five functional lines with one, three, two, three and four sublines, respectively (that is, 13 in total), which are primarily designed for fluorescence live imaging. Twelve of these sublines went through the procedure successfully, only the AGOC {ATub'H2B-mEmerald} #4 subline turned out to be heterozygous/homozygous sterile (*Supplementary file 6* for the mEmerald-labeled Lifeact-expressing sublines and *Supplementary file 7* for the mEmerald-labeled *beta-galactoside alpha-2,6-sialyltransferase 1*- and *histone H2B*-expressing sublines).

## Fluorescence live imaging of selected functional homozygous AGOC sublines

We performed long-term fluorescence live imaging of the embryonic development (*Strobl and Stelzer, 2016*) with three of the functional (mC/mC) homozygous sublines. We used a digital scanned laser light-sheet-based fluorescence microscope (*Keller et al., 2008*; *Keller and Stelzer, 2010*) in conjunction with previously published sample preparation protocols for *Tribolium* (*Strobl and Stelzer, 2014*; *Strobl et al., 2015*; *Strobl et al., 2017a*). The AGOC{Zen1'#O(LA)-mEmerald} #2 subline allows the characterization of actin dynamics within certain extra-embryonic membrane progenitor cells during gastrulation, visualizing the actomyosin cable that closes the serosa window (*Figure 4A* and *Video 1*). The AGOC{ARP5'#O(LA)-mEmerald} #1 subline provides strong fluorescence signal in the brain and ventral nerve cord and moderate signal throughout the

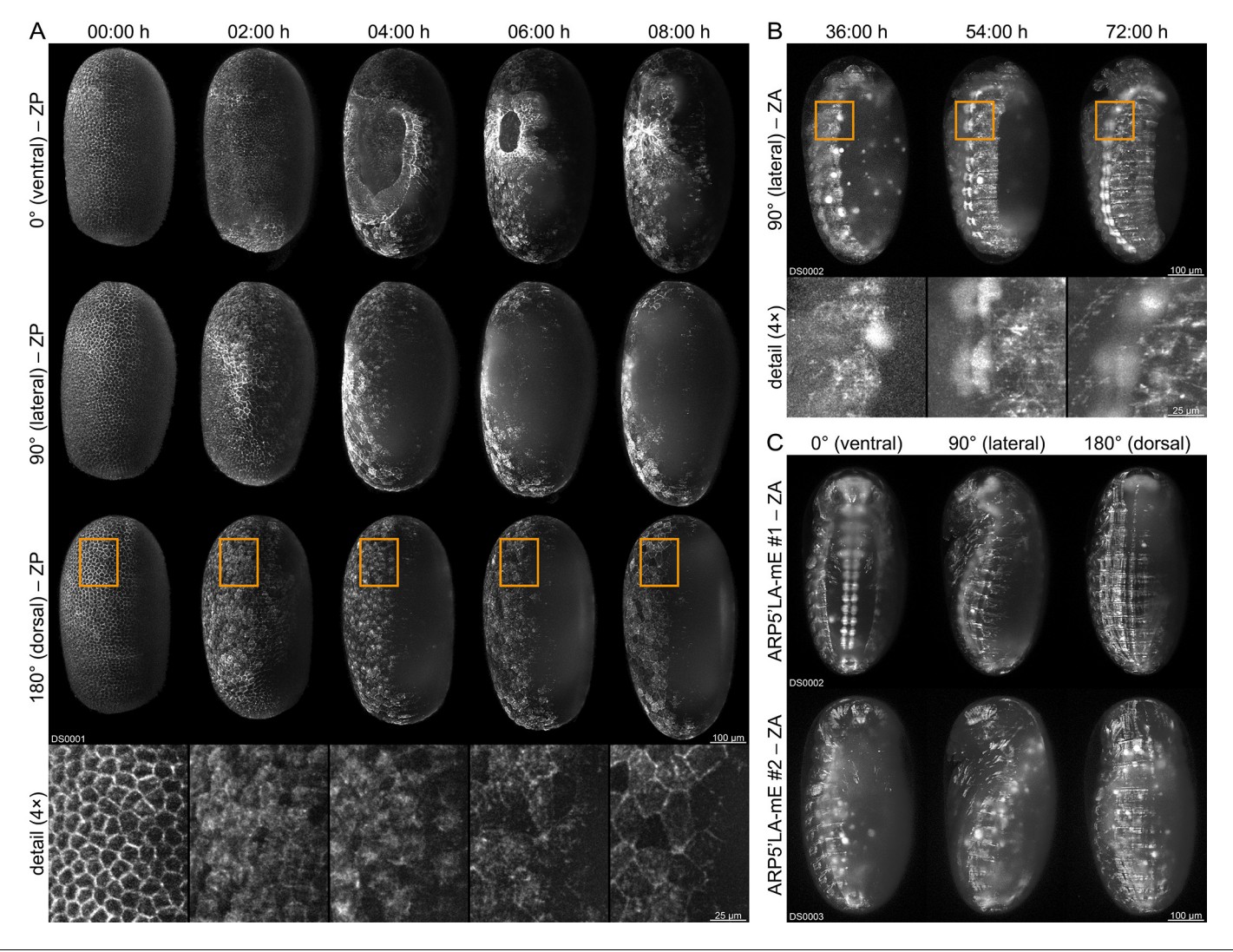

**Figure 4.** Fluorescence live imaging of selected functional (mC/mC) homozygous AGOC sublines. (**A**) An AGOC{Zen1'#O(LA)-mEmerald} #2 embryo during gastrulation. This subline permits the characterization of actin and actomyosin dynamics involved in serosa window closure (first row). It can also be used to describe the cytoskeleton rearrangement of the dorsal blastoderm cells (second row) and to analyze their change in appearance during differentiation to serosa cells (third row and enlarged images). (**B**) An AGOC{ARP5'#O(LA)-mEmerald} #1 embryo during germband retraction. In this subline, the brain and ventral nerve cord express mEmerald-labeled Lifeact on a high level, permitting the observation of neurulation. Enlarged images show the forming ganglia of the first and second thoracic segments. (**C**) Comparison of embryos from the AGOC{ARP5'#O(LA)-mEmerald} #1 and #2 sublines after dorsal closure. In contrast to the #2 subline, the fluorescence signal within the nervous system of the #1 subline is noticeably strong. ZP, Z maximum projection with image processing; ZA, Z maximum projection with intensity adjustment.

DOI: https://doi.org/10.7554/eLife.31677.016

remaining embryonic tissue (*Figure 4B* and *Video 2*). In contrast, the AGOC{ARP5'#O(LA)-mEmerald} #2 subline provides uniform fluorescence intensity during gastrulation, germband elongation, germband retraction and dorsal closure (*Figure 4C* and *Video 3*).

## Discussion

We explained the abstract genetic background of the AGOC vector concept and confirmed its straightforward applicability with *Tribolium*. The unique feature of our approach is that temporary ambiguities are avoided in any generation, since all genotypes are directly identified by specifically designed distinct phenotypes. Hence, AGOC-based workflows can be used to systematically create

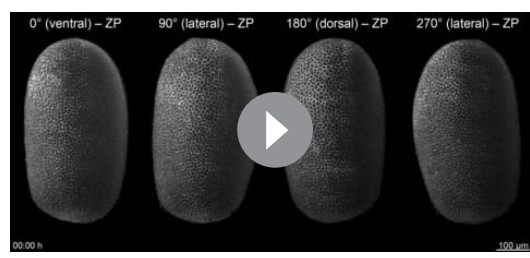

**Video 1.** Long-term live imaging of a (mC/mC) homozygous *Tribolium* embryo from the AGOC {Zen1'#O(LA)-mEmerald} #2 subline. Embryogenesis is shown along four directions from 00:00 hr to 24:00 hr with an interval of 00:30 hr between the time points. The video starts with the rearrangement of the blastoderm and ends during germband retraction. During gastrulation, the ventrally located serosa window is closed by a contracting actomyosin cable that separates the serosa and the amnion. Frame rate is five frames per second. ZP, Z maximum projection with image processing.
DOI: https://doi.org/10.7554/eLife.31677.017

progeny with relevant genotypes, as exemplified in this study for the creation of homozygous lines. Consequently, our concept provides many advantages that apply not only to *Tribolium* but also to many other model organisms: (i) Our approach saves manpower. For example, genotyping 30 to 40 *Tribolium* adults with genetic assays takes about one afternoon (*Strobl et al., 2017b*), while processing the same number of individuals with a stereo microscope takes less than ten minutes. (ii) The concept does not require any further consumables. (iii) When genetic assays are used, the 'slowest' member of a group defines the earliest convenient time point for synchronized genotyping, while our concept also supports unsynchronized genotyping of single organisms. (iv) Our approach is non-invasive and thus favorable when invasive procedures are incompatible with the experimental workflow. It can be performed even when sufficient amounts of genomic DNA cannot be obtained without severely injuring or even sacrificing the individual. (v) The concept simplifies transgenic organism handling since genotypes are determined directly. Quick and reliable

quantification, selection, mating and/or grouping of individuals can be performed during nearly all developmental stages. (vi) Our approach is less error-prone than genetic assays. In more than 300 independent instances, the progeny scores confirmed the phenotypically determined parental genotypes. (vii) Although homozygous transgenic lines can be systematically created with slightly less waiting time by using balancer chromosomes, a convenient number of balancer lines is only available for *Drosophila* (*Ashburner, 1989*). Furthermore, in the balancer-based approach, the insertion location has to be known, while our approach performs properly in random and semi-random insertion assays. (viii) Many special cases of transgenesis (four cases that occurred during our study are described within the Materials and methods section) can be explicitly identified and/or attended to. (ix) Specifically designed distinct phenotypes foster automation. For example, several approaches

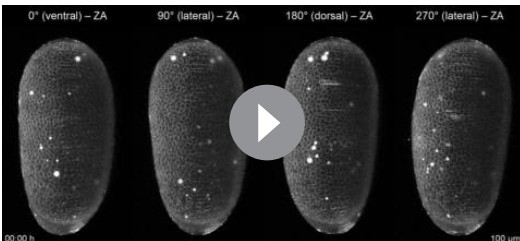

**Video 2.** Long-term live imaging of a (mC/mC) homozygous *Tribolium* embryo from the AGOC {ARP5'#O(LA)-mEmerald} #1 subline. Embryogenesis is shown along four directions from 00:00 hr to 96:00 hr with an interval of 00:30 hr between the time points. The video starts with the rearrangement of the blastoderm and ends after dorsal closure. This transgenic line exhibits strong fluorophore expression in the ventral nerve cord. Frame rate is five frames per second. ZA, Z maximum projection with image adjustment.
DOI: https://doi.org/10.7554/eLife.31677.018

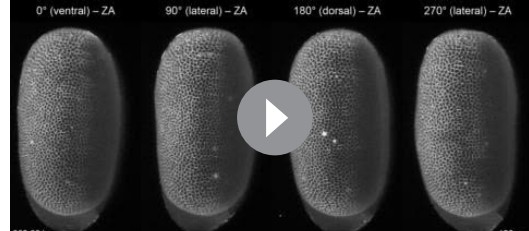

**Video 3.** Long-term live imaging of a (mC/mC) homozygous *Tribolium* embryo from the AGOC {ARP5'#O(LA)-mEmerald} #2 subline. Embryogenesis is shown along four directions from 00:00 hr to 120:00 hr with an interval of 00:30 hr between the time points. The video starts with the rearrangement of the blastoderm and ends after dorsal closure. In contrast to the #1 subline (*Video 2*), this subline does not exhibit strong fluorophore expression in the ventral nerve cord. Frame rate is five frames per second. ZA, maximum projection with image adjustment.
DOI: https://doi.org/10.7554/eLife.31677.019

for the computer-controlled allocation of zebrafish embryos to 96-well plates have been suggested (*Graf et al., 2011*; *Mandrell et al., 2012*). Automation devices, equipped with a phenotype-adapted detection unit, in our case fluorescence, can be used to sort organisms with different genotypes according to their markers.

The functionality of the AGOC vector concept was confirmed with *Tribolium*, but due to the universality of the Cre-Lox system, it should work in all diploid model organisms. These include various insects, zebrafish, rodents, and even plants. For many insect species, modifying the basic architecture of the vector is not necessary. It has been shown that both the piggyBac transposon system and the 3×P3 promoter function properly in *Drosophila melanogaster* (*Sarkar et al., 2006*), its close relative *Drosophila suzukii* (*Schetelig and Handler, 2013*) and many other dipterans (*Hediger et al., 2001*; *Warren et al., 2010*; *Caroti et al., 2015*), including epidemiologically relevant mosquito species such as *Aedes aegypti* (*Kokoza et al., 2001*) and *Aedes albopictus* (*Labbé et al., 2010*). This also applies to multiple lepidopterans such as *Bicyclus anynana* (*Marcus et al., 2004*) and *Bombyx mori* (*Thomas et al., 2002*), other coleopterans (*Kuwayama et al., 2006*) as well as some hymenopterans, for example the honeybee *Apis mellifera* (*Schulte et al., 2014*). For several other dipteran species, such as the African malaria mosquito *Anopheles gambiae* (*Grossman et al., 2001*) as well as several tephritid (*Handler and Harrell, 2001*; *Schetelig et al., 2009*; *Raphael et al., 2011*) and calliphorid (*Heinrich et al., 2002*; *Allen et al., 2004*) species, the marker cassettes have to be modified, for example by replacing the artificial 3×P3 promoter with the *Drosophila polyubiquitin* promoter. If fluorescence-based markers interfere with the experimental workflow, pigmentation-based markers can be used. Eye pigmentation markers are available for *Drosophila melanogaster* (*Adams and Sekelsky, 2002*) and *Tribolium castaneum* (*Lorenzen et al., 2002*), but require the appropriate background strain. Another convenient and apparently universal option for insects are *arylalkylamine-N-acetyl transferase*-based markers, which lighten pigmentation throughout the cuticle and thus can be detected without microscopes (*Osanai-Futahashi et al., 2012*). Although the piggyBac transposon system works properly in zebrafish (*Lobo et al., 2006*) and the 3×P3 promoter is believed to work in a broad variety of animal species (*Berghammer et al., 1999*), it may be convenient to transit to the well-established Tol2 transposon system (*Kawakami et al., 2000*) and to replace the 3×P3 promoter in the marker cassettes with endogenous alternatives, for example the eye-specific *cryaa* or the muscle-specific *503unc* promoter (*Berger and Currie, 2013*). For mouse, epidermal (*Ikawa et al., 1995*; *Zhu et al., 2005*) and eye-specific (*Cornett et al., 2011*) fluorescence-based as well as fur color-based (*Zheng et al., 1999*) markers have been established.

In this study, we used the AGOC vector concept in conjunction with transposon-mediated transgenesis to systematically create functional homozygous *Tribolium* lines that are primarily designed for fluorescence live imaging of embryonic development. However, our approach can also be used in insertional mutagenesis knock-out assays, independent of whether large-scale transposon-mediated remobilization with subsequent screening is performed (*Trauner et al., 2009*), or genes or genetic elements are specifically rendered inoperative, for example, by using genome engineering techniques such as CRISPR/Cas9, where AGOC-based transgenes can be integrated into targeted genomic locations via either homology-based repair or non-homologous end-joining (*Gilles and Averof, 2014*; *Gilles et al., 2015*).

Studies that utilize genetically manipulated organisms require researchers to rear their lines for many years. A certain numbers of individuals with known genotypes are required during this period, either to maintain the lines or to use them in experiments. This results in a total demand of hundreds to thousands of organisms. The AGOC vector concept supports well-designed experimental strategies in the following scenarios: (i) Transgenic lines that are, for example, specifically designed for fluorescence live imaging are easily maintained as homozygotes since continuous genotyping and/or curation are not necessary. However, the initial workload following transgenesis that is required to create homozygous lines can be very high and thus a limiting factor. As shown in this study, these efforts are significantly reduced by using the AGOC vector concept. (ii) In knock-out assays of genes that result in homozygous lethality, the respective lines are maintained as hemizygotes, which are usually viable and phenotypically inconspicuous. When two hemizygous organisms are mated, only one quarter of their progeny are homozygous for the knock-out, while half are hemizygous and one quarter resemble the wild type. Certain experimental approaches, for example, fluorescence live imaging or transcriptome/proteome analyses, require the researcher to commence with all descendants and to select the homozygous knock-out individuals as soon as discrimination is possible,

that is, when the phenotype manifests or when biological material for genetic assays can be obtained. By using our approach with appropriate markers, a preselection can be performed. This narrows the efforts down to relevant individuals and appropriate controls, for example, down to about one quarter of the currently required number. The AGOC vector concept, broadly adapted to established and emerging model organisms, contributes significantly to ethically motivated endeavors to minimize the number of wasted animals.

# Materials and methods

Key resources table

| Reagent type (species) or resource | Designation | Source or reference | Identifiers | Additional information |
|---|---|---|---|---|
| Gene (*Tribolium castaneum*) | *Tubulin alpha 1-like protein* (ATub) | PMID: 18625397 | Gene ID: 656649 | - |
| Gene (*T. castaneum*) | *Heat shock protein 68b* (HSP68) | PMID: 22890852 | Gene ID: 100142517 | - |
| Gene (*T. castaneum*) | *Zerknüllt 1* (Zen1) | This paper | Gene ID: 641533 | The *Tribolium* Zen1 promoter was not cloned previously |
| Gene (*T. castaneum*) | *Actin-related protein 5* (ARP5) | This paper | Gene ID: 655949 | The *Tribolium* ARP5 promoter was not cloned previously |
| Gene (*T. castaneum*) | *Histone H2B* (H2B) | This paper | Gene ID: 661713 | The *Tribolium* H2B open-reading frame was not cloned previously |
| Gene (*T. castaneum*) | *Beta-galactoside alpha-2,6-sialyltransferase 1* (SiaTr) | This paper | Gene ID: 657186 | The *Tribolium* SiaTr open-reading frame was not cloned previously |
| Strain, strain background (*T. castaneum*) | Plain-White-As-Snow (PWAS) background strain | PMID: 25555987 | - | New hybrid: pearl background strain that also carries the light ocular diaphragm mutation |
| Recombinant DNA reagent | pUC57[AGOC] | This paper | - | Gene synthesis (Genewiz) |
| Recombinant DNA reagent | pGS[#P'#O(LA)-mEmerald] | This paper | - | Gene synthesis (Invitrogen) |
| Recombinant DNA reagent | pGS[ACOS] | This paper | - | Gene synthesis (Invitrogen) |
| Recombinant DNA reagent | pBSII-IFP-CDS | PMID: 19716359 | - | Kind gift from Malcom Fraser (University of Notre Dame, Indiana, United States) |
| Recombinant DNA reagent | pTriEx-HTNC | PMID: 11904364 | Addgene ID: 13763 | Ordered from Addgene. |
| Recombinant DNA reagent | pGEM-T Easy | Promega | Catalog No: A1360 | - |

## *Tribolium castaneum* strains and rearing

For this study, a *T. castaneum* (NCBITaxon:7070) double mutant background strain was created, which carries the pearl (*Grubbs et al., 2015*) and light ocular diaphragm (*Mocelin and Stuart, 1996*) mutations that result in completely unpigmented eyes. This strain was called Plain-White-As-Snow (PWAS) and used as a donor for genomic DNA and messenger RNA as well as for the creation of transgenic lines. Cultures were kept in groups of 150–500 individuals on growth medium (full grain wheat flour (113061006, Demeter) supplemented with 5% (wt/wt) inactive dry yeast (62–106, Flystuff) in 1 l glass bottles in a 12:00 hr light/12:00 hr darkness cycle at 25°C and 70% relative humidity (DR-36VL, Percival Scientific).

## *Tribolium* genomic DNA/mRNA extraction and complementary DNA synthesis

Approximately 20 PWAS adults (40 mg) were starved for 24 hr before genomic DNA was extracted with the Blood and Tissue Kit (69504, Qiagen) according to the manufacturer's instructions. From approximately 10 adults (20 mg), messenger RNA was extracted with TRIzol Reagent (15596026, Thermo Fisher Scientific) and complementary DNA was transcribed with the Superscript III reverse transcriptase (12080093, Thermo Fisher Scientific) using random hexamer primers.

## Experimental design

For germline transformation, the piggyBac transposon system (*Handler, 2002*) was chosen, which is highly active in *Tribolium*. This study utilized a set of vectors based on in silico design and de novo synthesis. This section explains the architecture of the three most important vectors in general, while

the detailed molecular biological procedure is explained within the following sections. All intermediate and transformation-ready vectors used in this study are based on pAVOIAF{#1–#2–#3–#4} (*Figure 1—figure supplement 3*). Between the unique AatII and PciI sites, this vector carries a transposon cassette which consists of (i) the piggyBac 3' terminal repeat, (ii) a four-slot (#1 to #4) cloning site and (iii) the 5' piggyBac terminal repeat. The repeats have the minimal length (235 and 310 bp, respectively) necessary for efficient transposition (*Li et al., 2005*). The four-slot cloning site consists of four restriction enzyme site pairs, XmaI/SpeI for #1, HindIII/XbaI for #2, XhoI/NheI for #3 and AflII/AvrI for #4. The pairs are separated by 18 bp Phe-Arg-Glu-Asp-Asp-Tyr (FREDDY) spacers. For convenience, PmeI sites were placed at upstream and downstream of the four-slot cloning site as well as between the restriction enzyme site pairs.

In #3 and #4 of pAVOIAF{#1–#2–#3–#4}, mOrange- and mCherry-based eye-specific transformation markers (mO and mC, respectively) were inserted (in reverse orientation) that consist of (i) the artificial 3×P3 promoter (*Berghammer et al., 1999*), (ii) the codon-optimized open-reading frames for the respective fluorescent protein, that is, mOrange2 (*Shaner et al., 2008*) or mCherry (*Shaner et al., 2004*) and (iii) the SV40 poly(A) (*van den Hoff et al., 1993*). Both markers are flanked by incompatible Lox sites (the spacer is underlined, deviations are marked bold): upstream by a LoxP (5'-ATAACTTCGTATAG**C**ATAC**A**TTATACGAAGTTAT-3') and downstream by a LoxN (5'-ATAACTTCGTATAG**T**ATAC**C**TTATACGAAGTTAT-3') site. For convenience, the resulting vector was termed pAGOC (*Figure 1—figure supplement 1*).

In #2 of pAGOC, a modular fluorescent protein expression cassette was inserted, which consists of (i) a two-slot cloning site composed of a promoter (#P) and an open-reading frame (#O) slot, (ii) a 9 bp Ala-Ala-Ala linker, (iii) the codon-optimized mEmerald open-reading frame (*Tsien, 1998*) and (iv) an elongated variant of the SV40 poly(A) (*van den Hoff et al., 1993*). The #P slot can be accessed by the AscI/FseI site pair, or alternatively scarlessly by the double BtgZI site pair. The #O slot carries the open-reading frame for the *Saccharomyces cerevisiae* Lifeact peptide tag (*Riedl et al., 2008*) per default and can be accessed by the FseI/NotI site pair. The mEmerald open-reading frame can be accessed by the NotI/SbfI site pair. For convenience, the resulting vector was termed pAGOC{#P'#O(LA)-mEmerald} (*Figure 1—figure supplement 2*). Any exogenous or endogenous promoter can be inserted in the #P slot to generate the spatiotemporal activity pattern of choice. For the #O slot, the Lifeact open-reading frame can be replaced with any open-reading frame to change the subcellular localization of the fluorescent protein. In the default configuration, the Lifeact peptide tag will guide mEmerald to the actin cytoskeleton.

## Molecular biology

In this study, 25 vectors (*Figure 1—figure supplement 4* and *Supplementary file 8*) plus the commercial subcloning vector pGEM-T Easy (A1360, Promega) were used. Three were ordered as gene synthesis plasmids, two were previously published and obtained from the respective laboratories or from Addgene, six are library vectors, while the remaining 13 vectors are derivates. For all PCRs, Phusion High Fidelity DNA polymerase (M0530L, New England BioLabs) was used, and T4 DNA ligase (M0202L, New England BioLabs or provided with the pGEM-T Easy vector) for all ligations. Cloning primers are listed in *Supplementary file 9*.

## Molecular biology: the promoter and open-reading frame library vectors

The respective promoter and open-reading frame sequences were amplified from genomic or complementary DNA by using the appropriate extraction PCR primer pairs (C1 for *tubulin alpha 1-like protein* (ATub'), C2 for *zerknüllt 1* (Zen1'), C3 for *actin-related protein 5* (ARP5') and C4 for *heat shock protein 68b* (HSP68') as well as C5 for *beta-galactoside alpha-2,6-sialyltransferase 1* transcription variant X1 ('SiaTr) and C5 for *histone H2B* ('H2B)). Amplification was followed by A-tailing using the Recombinant Taq DNA polymerase (10342020, Thermo Fisher Scientific) and ligation into pGEM-T Easy. The resulting vectors were termed pTC-ATub'-GEM-T Easy, pTC-Zen1'-GEM-T Easy, pTC-ARP5'-GEM-T Easy, pTC-HSP68'-GEM-T Easy, pTC-'SiaTr-GEM-T Easy and pTC-'H2B-GEM-T Easy. To create the hybrid promoter/open-reading frame library vectors, the sequences were amplified from the library vectors or pTriEx-HTNC (*Peitz et al., 2002*) with the respective fusion PCR primer pairs (either C7 for HSP68' / 'NLS-Cre, or C8 for ATub' / 'H2B) and fused in a secondary PCR

reaction using both PCR products as a template and the promoter forward primer (C7-1 or C8-1, respectively) and the open-reading frame reverse primer (C7-4 or C8-4, respectively). The primer pairs introduce upstream an AscI and downstream a NotI site or upstream a NheI and downstream a XhoI site, respectively. The fusion PCR products were inserted into pGEM-T Easy as described above. The resulting vectors were termed pTC-ATub'H2B-GEM-T Easy and pTC-HSP68'NLS-Cre-GEM-T Easy.

## Molecular biology: the pUC[AGOC] and pAGOC vectors

A hybrid sequence, consisting of (i) the transposon cassette as well as (ii) mO and mC and their flanking Lox sites in #3 and #4 as described above, was de novo synthetized and inserted into the unique NdeI and PstI sites of pUC57-Kan (GeneBank accession number JF826242.2). The resulting vector was termed pUC57[AGOC]. The insert was PCR amplified with primer pair C9, which introduced upstream an AatII and downstream a PciI site. The PCR product and pUC57[AGOC] were digested accordingly, and the insert was reintegrated into the vector, removing 629 functionless bp. The resulting vector was termed pAGOC and used (i) as a transformation-ready vector for germline transformation, and (ii) as an intermediate vector for further cloning operations.

## Molecular biology: the pGS[#P'#O(LA)-mEmerald] and pAGOC{#P'#O (LA)-mEmerald} vectors

A hybrid sequence, consisting of (i) a HindIII site, (ii) the modular fluorescent protein expression cassette as described above and (iii) a XbaI site, was de novo synthetized and inserted into the unique SfiI site of pMK-RQ (Thermo Fisher Scientific). The resulting vector was termed pGS[#P'#O(LA)-mEmerald]. The insert was excised from the backbone with HindIII/XbaI and inserted into #3 of the pAGOC vector. The resulting vector was termed pAGOC{#P'#O(LA)-mEmerald} and used as an intermediate vector for further cloning operations.

## Molecular biology: the pAGOC{ATub'#O(LA)-mEmerald}, pAGOC {Zen1'#O(LA)-mEmerald} and pAGOC{ARP5'#O(LA)-mEmerald} vectors

The respective promoter sequences were amplified from the library vectors with the respective primer pairs (C10 for ATub', C11 for Zen1' and C12 for ARP5'), which introduced upstream an AscI and downstream a BsmBI (ATub') or BsaI (Zen1' and ARP5') site. The PCR products were digested accordingly, and the pAGOC{#P'#O(LA)-mEmerald} vector was digested with BtgZI, which led to compatible overhangs and allowed scarless insertion of the promoter sequences into #P. The resulting vectors were termed pAGOC{ATub'#O(LA)-mEmerald}, pAGOC{Zen1'#O(LA)-mEmerald} and pAGOC{ARP5'#O(LA)-mEmerald} and were used for germline transformation.

## Molecular biology: the pAGOC{#P'SiaTr-mEmerald} and pAGOC {ATub'SiaTr-mEmerald} vectors

The 'SiaTr open-reading frame sequence was amplified from the pTC-'SiaTr-GEM-T Easy vector with primer pair C13, which introduced upstream an FseI and downstream a NotI site. The PCR product and the pAGOC{#P'#O(LA)-mEmerald} vector were digested accordingly and the insert was inserted into #O of the vector. The resulting vector was termed pAGOC{#P'SiaTr-mEmerald} and used as an intermediate vector for further cloning operations. The *tubulin alpha 1-like protein* promoter sequence was amplified from the pTC-ATub'-GEM-T Easy vector with primer pair C10, which introduced upstream an AscI and downstream a BsmBI site. The PCR product was digested accordingly, and the pAGOC{#P'SiaTr-mEmerald} vector was digested with BtgZI, which led to compatible overhangs and allowed scarless insertion of the promoter sequence into #P of the intermediate vector. The resulting vector was termed pAGOC{ATub'SiaTr-mEmerald} and used for germline transformation.

## Molecular biology: the pAGOC{ATub'H2B-mEmerald} vector

The ATub'H2B promoter/open-reading frame sequence was excised from pTC-ATub'H2B-GEM-T Easy with AscI and NotI and inserted into #P'#O of the accordingly digested pAGOC{#P'#O(LA)-mEmerald} vector. The resulting vector was termed pAGOC{ATub'H2B-mEmerald} and used for germline transformation.

## Molecular biology: the pAVOIAF{#1–#2–HSP68'NLS-Cre–mC}, pGS[ACOS] and pICE{HSP68'NLS-Cre} vectors

The HSP68'NLS-Cre recombinase promoter/open-reading frame sequence was excised from pTC-HSP68'H2B-GEM-T Easy with NheI and XhoI and inserted (in reverse orientation) into #3 of the accordingly digested pAGOC vector, replacing mO and the flanking Lox sites. The resulting vector was termed pAVOIAF{#1–#2–HSP68'NLS-Cre–mC} and used as an intermediate vector for further cloning operations. A hybrid sequence, which consists (beside other elements) of the mCerulean-based eye-specific transformation marker (mCe) that is composed of (i) the artificial 3×P3 promoter, (ii) the codon-optimized open-reading frame for mCerulean2 (*Markwardt et al., 2011*) and (iii) the SV40 poly(A), was de novo synthetized and inserted into the unique SfiI site of pMK-RQ (Thermo Fisher Scientific). The resulting vector was termed pGS[ACOS]. Next, mCe was amplified with primer pair C14, which introduced upstream an AflII and downstream an AvrII site. The PCR product was digested accordingly and inserted into #4 of pAVOIAF{#1–#2–HSP68'NLS-Cre–mC}, replacing mC and the flanking Lox sites. The resulting vector was termed pICE{HSP68'NLS-Cre} and used for germline transformation to create the Cre recombinase-expressing helper lines.

## Molecular biology: the pATub'piggyBac vector

The ATub promoter and the piggyBac open-reading frame fragment were amplified from pTC-ATub'-GEM-T Easy and pBSII-IFP-ORF (*Yoshida et al., 2009*) with the C15 primer pairs and fused together in a secondary PCR reaction using both PCR products as a template as well as the promoter forward primer (C15-1) and the open-reading frame reverse primer (C15-4). The primers introduced upstream a SalI and downstream a BglII site, respectively. The fusion PCR product was digested and then reintegrated into the accordingly digested pBSII-IFP-ORF vector. The resulting vector was termed pATub'piggyBac and used as the transposase-expressing helper vector during germline transformation.

## Germline transformation

Approximately 500 F0 PWAS adults were incubated on 405 fine wheat flour (113061036, Demeter, Darmstadt, Germany) supplemented with 5% (wt/wt) inactive dry yeast (62–106, Flystuff, San Diego, CA) at 25°C and 70% relative humidity in light for 2 hr. After the incubation period, the adults were removed and the embryos (around 700 to 900) were extracted from the flour and incubated another hour as stated above. Next, the embryos were briefly washed in 10% (vol/vol) sodium hypochlorite (425044–250 ML, Sigma Adlrich) in autoclaved tap water for 10 s, stored in autoclaved tap water and lined up on microscopy slides within the next hour. The embryos were injected with a mixture of 500 ng/µl transformation-ready vector and 400 ng/µl pATub'piggyBac in injection buffer (5 mM KCl, 1 mM KH$_2$PO$_4$ in ddH$_2$O, pH 8.8). For injection, a microinjector (FemtoJet, Eppendorf) and 0.7 µm outer diameter capillaries (Femtotips II, Eppendorf) with an injection pressure of 400–800 hPa were used. After injection, the microscopy slides with embryos were placed on a 5 mm high 1% (wt/vol) broad range agarose (T846.3, Carl Roth) in tap water 'platform' within Petri dishes and incubated at 32°C. After 3 days, hatched larvae, that is, F1 potential mosaics, were collected and raised individually in single wells of 24-well plates as described above. Germline transformation resulted in a total of seven lines with 21 sublines, which are summarized in *Supplementary file 10*.

## Mating procedure, insert number determination cross and homozygous viability cross

All crossings were performed with single female-male pairs in small glass vials filled with 1.5 g or 2.5 g (F4 cross) of growth medium. Progeny were placed individually in wells of 24-well plates and scored for the presence of markers during pupal or adult stage by using a fluorescence stereo microscope (SteREO Discovery.V8, Zeiss) with appropriate filter sets (*Supplementary file 11*). For each pair, images in the reflected light and fluorescence channels were taken in parallel with appropriate controls. The mating procedure is described within the results section. A one-sample/two tailed Student's t-test was performed to determine whether the arithmetic means differ significantly from the theoretical Mendelian ratios. Insert numbers were determined by mating F2 hemizygotes with wild types and scoring the progeny, whereas a transgene distribution of 60% or less was interpreted as a single insertion. Homozygous viability was determined by mating two F3 hemizygotes and

scoring the progeny, whereas a transgene distribution of 70% or more was interpreted as a homozygous viable line.

## The AGOC vector concept in special cases of transgenesis

During the experimental validation of the AGOC vector concept, four special cases of transgenesis occurred and were attended to as follows: (i) The homozygous viability crosses indicated that the transgenes of the proof-of-principle AGOC #3, the functional AGOC{ATub'#O(LA)-mEmerald} #1 and the functional AGOC{ATub'H2B-mEmerald} #4 sublines are heterozygous/homozygous lethal (*Supplementary file 2*). However, F6 (mO/mC) heterozygotes were obtained for all three lines and the mating procedure could be performed successfully with the AGOC #3 and AGOC{ATub'#O(LA)-mEmerald} #1 sublines (*Table 1*, *Supplementary file 6* and *Supplementary file 7*). The mating procedure only aborted for the AGOC{ATub'#O(LA)-mEmerald} #4 subline, because the F6 (mO/mC) heterozygotes were sterile. Since the transgene of the proof-of-principle AGOC #3 subline might interfere with the *sialin-like* gene due to its insertion location (*Supplementary file 3*), and since both functional sublines mentioned above display a strong green fluorescence signal, hemizygous individuals of those three sublines are believed to be exposed to high stress levels, and that these levels are even higher in homozygotes. Thus, these transgenic sublines are believed to be essentially viable, but a certain percentage of the descendants fail to develop properly, which results in biased progeny ratios in the homozygous viability cross. (ii) In contrast to heterozygous/homozygous lethality, heterozygous/homozygous sterility cannot be estimated from the homozygous viability crosses. The AGOC{ATub'H2B-mEmerald} #4 subline was assumed to be heterozygous/homozygous lethal (*Supplementary file 2*), but mating a F5 (mO) post-recombination hemizygous female with a F5 (mC) post-recombination hemizygous male sibling resulted in F6 (mO/mC) heterozygotes. However, by mating F6 (mO/mC) heterozygous females with genotypically identical F6 males, no progeny was obtained (n = 12). Further, crossing F6 (mO/mC) heterozygous females and wild-type males did not result in any progeny (n = 12). To confirm sterility of both genders, F6 (mO/mC) heterozygous males were mated with wild-type females, which also did not result in any progeny (n = 8). (iii) The AGOC {Zen1'#O(LA)-mEmerald} #2 subline carries the transgene not on one of the nine autosomes, but on the X allosome. Thus, a slightly modified mating procedure was necessary to obtain F7 (mO/mO) and (mC/mC) homozygotes (*Figure 2—figure supplement 2*). (iv) The piggyBac transposon system is highly efficient in *Tribolium*, which results to a certain degree also from the 4 bp TTAA target sequence. However, due to this very short length of the targeting sequence, a certain probability for nested insertions is given, that is, the transgene inserts into another transformation vector, since these vectors also carry multiple TTAA target sequences. In the AGOC{Zen1'#O(LA)-mEmerald} #3 subline, an insertion of the transgene into the backbone of another vector occurred, and the nested transgene was subsequently inserted into the genome, as revealed by sequencing of the insertion junction (*Supplementary file 3*). This rare and undesired case is 'corrected' during the mating procedure of the AGOC vector concept, since within the F4 (mCe; mO-mC) double hemizygous generation, the Cre recombinase excises nearly one 'stitched equivalent' of the initial transformation vector from the genome. The F5 (mO) and (mC) hemizygous progeny then carries only one, but complete copy of the transgene.

One of the most obvious special cases of transgenesis, heterozygous/homozygous lethality, did not occur. However, the AGOC vector concept would allow the determination of this special case with a high degree of certainty. When F5 (mO) post-recombination hemizygous females are mated with F5 (mC) post-recombination hemizygous male siblings and no F6 (mO/mC) heterozygotes are obtained, it can be assumed that the transgene is heterozygous/homozygous lethal. Multiple other transgenesis special cases are possible (e.g. female- or male-only heterozygous/homozygous lethality or sterility, nested insertions of transgenes into transgenes, multiple inserts in close proximity on the same chromosome), but are not discussed here due to the very low probability of their occurrence.

## Determination of insertion junction via inverse PCR

To determine the insertion junction of the piggyBac-based transgene, the inverse PCR approach was chosen (*Triglia et al., 1988*; *Ochman et al., 1988*). All inverse PCR primers are listed in *Supplementary file 9*. At first, inverse PCR was performed for the junction at the 5' piggyBac

terminal repeat with the I-5' primer pair with up to eight different restriction enzymes, and if unsuccessful, also at the 3' piggyBac terminal repeat with the I-3' primer pair with up to six different restriction enzymes. PCR products were extracted from the gel, A-tailed, ligated into pGEM-T Easy and sequenced. For each successful inverse PCR, a control PCR at the respective other side was performed. For the control PCR, location-specific primers were used to perform a standard PCR. Inverse PCR was successful for 10 out of 19 lines, for 4 out of 6 proof-of-principle lines and 6 out of 13 functional lines. The sequencing results were aligned to the *Tribolium* genome (*Richards et al., 2008*) via the BeetleBase (*Wang et al., 2007*; *Kim et al., 2010*) (RRID:SCR_001955) BLAST. Insertion junctions are listed in *Supplementary file 3*.

### Light-sheet-based fluorescence microscopy

Long-term live imaging was performed with digitally scanned laser light-sheet-based fluorescence microscopy (DSLM, LSFM) (*Keller et al., 2008*; *Keller and Stelzer, 2010*) as described previously for *Tribolium* (*Strobl and Stelzer, 2014*; *Strobl et al., 2015*). In brief, embryo collection was performed with the F7+ continuative (mC/mC) homozygous lines for 1 hr at 25°C, and embryos were incubated for 15 hr at 25°C. Sample preparation took approximately 1 hr at room temperature (23 ± 1°C), so that embryos were at the beginning of gastrulation. Embryos were recorded along four pair-wise orthogonal directions, that is, in the orientations 0°, 90°, 180° and 270°, with an interval of 30 min. All shown embryos survived the imaging procedure, developed to healthy and fertile adults, and when mated with wild types, produced only transgenic progeny that were also fertile. Metadata for the three datasets is provided in *Supplementary file 12*.

## Acknowledgements

We thank J Alexander Ross, Katharina Hötte, Berit Reinhardt and Sigrun Becker for their valuable support. The pBSII-IFP-ORF vector was a kind gift from Malcom Fraser (University of Notre Dame, Indiana, United States).

## Additional information

### Competing interests

Frederic Strobl, Ernst HK Stelzer: Patent application for the vector concept (DE file number: 10 2017 112 863.8). The other author declares that no competing interests exist.

### Funding

| Funder | Grant reference number | Author |
| --- | --- | --- |
| Deutsche Forschungsge-meinschaft | CEF-MC, EXC 115 | Frederic Strobl<br>Anita Anderl<br>Ernst HK Stelzer |

The funders had no role in study design, data collection and interpretation, or the decision to submit the work for publication.

### Author contributions

Frederic Strobl, Conceptualization, Data curation, Formal analysis, Supervision, Validation, Investigation, Visualization, Methodology, Writing—original draft, Project administration, Writing—review and editing; Anita Anderl, Investigation, Methodology, Writing—review and editing; Ernst HK Stelzer, Conceptualization, Data curation, Formal analysis, Supervision, Funding acquisition, Validation, Writing—original draft, Project administration, Writing—review and editing

### Author ORCIDs

Ernst HK Stelzer https://orcid.org/0000-0003-1545-0736

**Decision letter and Author response**
Decision letter https://doi.org/10.7554/eLife.31677.041
Author response https://doi.org/10.7554/eLife.31677.042

## Additional files

**Supplementary files**

• Supplementary file 1. F2 insert number determination cross. F2 (mO-mC) founder females were mated with wild-type males and the progeny were scored. Segregation of 60% or fewer transgenic descendants was defined as the criterion for one insert. No deviators could be identified.
DOI: https://doi.org/10.7554/eLife.31677.020

• Supplementary file 2. F3 homozygous viability crosses. Two F3 (mO-mC) pre-recombination hemizygous siblings were mated and the progeny were scored. Segregation of 70% or more transgenic descendants was defined as the criterion for homozygous viability. Deviators are marked bold.
DOI: https://doi.org/10.7554/eLife.31677.021

• Supplementary file 3. Insertion junctions. In the Junction column, the piggyBac TTAA insertion/excision target sequence is marked bold.
DOI: https://doi.org/10.7554/eLife.31677.022

• Supplementary file 4. Generations. In this table, the F0 to F7 and their characteristics are summarized.
DOI: https://doi.org/10.7554/eLife.31677.023

• Supplementary file 5. Mating procedure results for the two proof-of-principle AGOC #5 and #6 sublines from the F3 to the F7 generation with swapped genders as well as with an alternative Cre-expressing homozygous helper subline, ICE{HSP68'NLS-Cre} #2. Bold entries mark progeny that were used in the subsequent cross. F6-S, F7-O and F7-C are control crosses. No significant differences between the arithmetic means and the theoretical Mendelian ratios were found. See Source Data 1 for raw scores ordered by transgenic sublines.
DOI: https://doi.org/10.7554/eLife.31677.024

• Supplementary file 6. Mating procedure results for six of the thirteen functional AGOC sublines (Lifeact only) from the F3 to the F7 generation. Bold entries mark progeny that were used in the subsequent cross. F6-S, F7-O and F7-C are control crosses. No significant differences between the arithmetic means and the theoretical Mendelian ratios were found. See Source Data 1 for raw scores ordered by transgenic sublines.
DOI: https://doi.org/10.7554/eLife.31677.025

• Supplementary file 7. Mating procedure results for seven of the thirteen functional AGOC sublines (Non-Lifeact) from the F3 to the F7 generation. Bold entries mark progeny that were used in the subsequent cross. F6-S, F7-O and F7-C are control crosses. No significant differences between the arithmetic means and the theoretical Mendelian ratios were found. See Source Data 1 for raw scores ordered by transgenic sublines.
DOI: https://doi.org/10.7554/eLife.31677.026

• Supplementary file 8. Vector summary. The 24 vectors used/created in this study listed in order of their type. Numbers in square brackets within the Source/molecular cloning column refer to the respective entry. See also (*Figure 1—figure supplement 4*).
DOI: https://doi.org/10.7554/eLife.31677.027

• Supplementary file 9. Cloning and inverse PCR primer pairs. Primer pairs are listed in order of appearance in the Materials and methods section and *Supplementary file 3*. The Applied Biosciences web calculator (www6.appliedbiosystems.com/support/techtools/calc) was used to calculate the melting temperature $T_M$. In case of primers with overhangs, the $T_M$ was only calculated for the annealing part. Primer that introduce a restriction enzyme site also carry a 6 bp (5'-AAATTT-3') buffer at the 5' end. Several primers have been used in multiple inverse PCRs and are therefore also listed multiple times, as annotated within the Comment column. ExPCR, extraction polymerase chain reaction; SiRePCR, size reduction polymerase chain reaction; FuPCR, fusion polymerase chain reaction; TrPCR, transfer polymerase chain reaction; InvPCR, inverse polymerase chain reaction; ConPCR, control polymerase chain reaction; FD, forward; RV, reverse.

DOI: https://doi.org/10.7554/eLife.31677.028

• Supplementary file 10. Transgenic lines and sublines. In total, 7 transgenic lines with 21 sublines were created, that is, 6 proof-of-principle AGOC sublines, 13 functional AGOC sublines and 2 helper sublines. Two of the functional AGOC sublines have been analyzed with live imaging previously, live imaging data for three more is provided in this study.
DOI: https://doi.org/10.7554/eLife.31677.029

• Supplementary file 11. Fluorescence stereo microscope filter sets. All components were obtained from AHF Analysentechnik, Tübingen, Germany.
DOI: https://doi.org/10.7554/eLife.31677.030

• Supplementary file 12. Metadata and parameter for the long-term live-imaging datasets DS0001-0003.
DOI: https://doi.org/10.7554/eLife.31677.031

• Supplementary file 13. All 25 vector sequences as Genebank (.gb) and Geneious (.geneious) files compressed into a single zipped folder (.zip).
DOI: https://doi.org/10.7554/eLife.31677.032

• Transparent reporting form
DOI: https://doi.org/10.7554/eLife.31677.033

## Major datasets

The following datasets were generated:

| Author(s) | Year | Dataset title | Dataset URL | Database, license, and accessibility information |
|---|---|---|---|---|
| Strobl F, Anderl A, Stelzer EHK | 2018 | Strobl2018A-DS0001: AGOC {Zen1'#O(LA)-mEmerald} #2 long-term live imaging data acquired with light-sheet-based fluorescence microscopy | https://doi.org/10.5281/zenodo.1193297 | Publicly available at Zenodo (https://zenodo.org/) |
| Strobl F, Anderl A, Stelzer EHK | 2018 | Strobl2018A-DS0002: AGOC {ARP5'#O(LA)-mEmerald} #1 long-term live imaging data acquired with light-sheet-based fluorescence microscopy | https://doi.org/10.5281/zenodo.1194027 | Publicly available at Zenodo (https://zenodo.org/) |
| Strobl F, Anderl A, Stelzer EHK | 2018 | Strobl2018A-DS0003: AGOC {ARP5'#O(LA)-mEmerald} #2 long-term live imaging data acquired with light-sheet-based fluorescence microscopy | https://doi.org/10.5281/zenodo.1194029 | Publicly available at Zenodo (https://zenodo.org/) |

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
