## [Decision Letter]

Thank you for submitting your article "A universal vector concept for a direct genotyping of transgenic organisms and a systematic creation of homozygous lines" for consideration by *eLife*. Your article has been favorably evaluated by K VijayRaghavan (Senior Editor) and three reviewers, one of whom is a member of our Board of Reviewing Editors. The following individual involved in review of your submission has agreed to reveal her identity: Susan Brown (Reviewer #2).

The reviewers have discussed the reviews with one another and the Reviewing Editor has drafted this decision to help you prepare a revised submission.

I have attached the reviews in full below, with the hope that you will be able to address essentially all of the requested revisions. Overall, the reviewers appreciated the content of the paper, but felt that it needs significant revision to make it clearer, more concise, and more accessible to a broad readership. The revision thus should have simpler figures, less supplementary data, and more concise Results and Discussion sections. I think it will also be important to revise the rationale, putting your method's strengths and weaknesses better in context with existing methods for selecting and working with mutants and transgenics. Reviewer #3 had some important contents in this area. I look forward to seeing your revision.

*Reviewer #1:*

In this paper, Strobl et al. present a germline transformation protocol using a novel vector they've developed, that streamlines the process of making a transgene insert into the genome homozygous. The concept is simple yet novel and applicable, with modifications, to a wide range of animal genetic models that lack other means of easily homozygosing transgenes, such as by the use of balancers or targeted insertion sites. The authors demonstrate their technique using vectors they've constructed and fluorescent eye markers in the flour beetle, *Tribolium*. The technique works, and would appear to be a useful one for this beetle and other model genetic systems that lack advanced genetic tools. Overall the paper is well put together; it's complete and easy to understand and includes the tests needed to validate the method presented. However the paper's methods and figures sections are much longer and more detailed than they need to be, so I suggest simplifying these to make the whole more accessible to the interested reader. Specific comments follow below:

1) Subsection “Systematic creation of homozygous lines”, first paragraph. To put things in perspective, this would take 3 generations in *Drosophila* using balancers, so a 4 generation procedure in *Tribolium* is not revolutionary, but rather an incremental advance.

2) The complete DNA sequences of the vectors should be made available as supplemental data for this paper. This is quite important to make the method accessible to the scientific community.

3) Would it not make sense to invert one of the marker genes in the vector to prevent read-through? Please comment.

4) In Figure 2, why is the first box F3 rather than F1 or F2? Please comment. Why is there a dotted line connecting rows 3 and 4, rather than solid?

5) Figure 3 and Figure 4 are much more complex than necessary, and they didn't reproduce well. Please simplify these two figures.

6) The text in Table 1 is too small to read. Could this be presented in a more intuitive form?

7) The supplementary data are longer and more complex than necessary. Please include only what is necessary to convey the method to the reader.

8) In the Discussion, please include a list of the organisms in which this technique might be useful, and discuss how the vector would have to be modified to be used in these other organisms. Discussing these issues is important to ensure adoption of this method by the broader research community.

*Reviewer #2:*

The authors describe important technical improvements to transgenic vector constructs that allow efficient production of homozygous transgenic strains. Their concept is applicable to any system in which one wishes to generate transgenic strains. The methodology is well written in the overall sense, but lacks the clarity that correct genetic terminology can bring. In the detailed comments I provide, I have suggested corrections for the first time I encountered an issue. However, it they should be applied throughout the narrative and legends.

Impact statement:

"allow to create homozygous transgenic lines systematically" is awkward, may I suggest:

“allow us to systematically generate homozygous transgenic lines…”

Or

“allow the systematic construction of homozygous transgenic lines…”

Main text, first paragraph: “crossing setup[…]”; Do you mean, “crossing scheme”?

“allow to distinguish” is awkward, suggestion:

“allow one to distinguish”.

“purposely produced” is awkward, suggestion: “specifically designed”.

“phenotypically clearly distinguishable” – I understand the need for emphasis, but two adverbs seem like a little much.

Clearly distinguishable is much smoother and just as informative. After all aren't all transformation markers producing a phenotype?

"of both markers" a little clearer is "of the two markers".

"The progeny was" – throughout the paper you are using progeny in the plural (you could substitute children, not child). Thus it should be progeny were.

"which contains a mOrange-based6 and a mCherry-based7 eye-specific8 transformation marker". Suggestion: “That contains both mOrange-based and mCherry-based eye-specific transformation markers"

“appropriate excitation and emission”. Suggestion: “appropriate excitation and emission filters”.

"which were outcrossed against wild-type males" a little smoother and clearer "and these were crossed to".

Materials and methods:

“study utilizes a setup of vectors that are based on in silico designed and de novo synthetized sequences”. Suggestion: “study utilizes a set of vectors based on in silico design and de novo synthesis.”

“sublines were termed AGOC” – suggestion: “designated” or “called”

“Until this step, our route did not differ from most standard transgenic organisms establishment procedures.” A little smoother and clearer: “Through this step, our scheme did not differ from most standard procedures to generate transgenic organisms.”

"which carry mO and mC consecutively on the[…]" change to "that carry mO and mC in tandem".

"resulting in F4 mCe x mO-mC double hemizygotes in which Cre-mediated recombination occurs (Table 1, 'F3' row)" – this phrase actually follows from the previous sentence not the one to which it is attached. The genotype nomenclature is misleading, how about a completely new sentence:

“In the resulting F4 double hemizygotes (Ce; mO-mC), Cre-mediated recombination occurs (Table 1, 'F3' row).” Also please note the cross occurs in the F3, the F4 should be described (Ce; mO-mC) not mCe x mO-mC). Please use a semicolon to denote non-linkage.

Subsection “Systematic creation of homozygous lines”, same as above, the following is a cross not a genotype mCe x mO-mC. And "were outcrossed against wild-type males" Why are outcross and cross used interchangeably? It would be good to be strictly consistent in terminology usage.

Also, inbreeding is not the same as selfing. (Beetles cannot be selfed, since they are not hermaphrodites.) The naïve reader will appreciate the clarity if you check through entire manuscript for term usage consistency.

"F5 mO-only post-recombination hemizygous females were brother-sister crossed against F5 mC-only post recombination hemizygous males (Figure 2 and Figure 3, third row), resulting in F6 mO/mC heterozygotes, which carry once again both markers (Table 1, 'F5' row).” Since you are going to go into detail re cis/trans relationship of transgenes, why not use that terminology? May I suggest: “F5 mO-only post-recombination hemizygous females were mated to mC-only post recombination hemizygous male siblings (Figure 2 and Figure 3, third row), resulting in F6 mO/mC heterozygotes that carry both markers once again (Table 1, 'F5' row), but now in trans.”

"In contrast to the F3 mO-mC pre-recombination hemizygotes, which show the same phenotype but carry both markers consecutively on the maternal chromosome, the F6 mO/mC heterozygotes carry mO on the maternal and mC on the paternal chromosome. This was proven by crossing F6 mO/mC heterozygous females against wild-type males (Table 1, 'F6-S' row)." Suggestion: “In contrast to the F3 mO-mC pre-recombination hemizygotes, which display the same phenotype but carry both markers in tandem on the maternal chromosome, the F6 mO/mC heterozygotes carry mO on the maternal and mC on the paternal chromosome. This was demonstrated by crossing F6 mO/mC heterozygous females against wild-type males (Table 1, 'F6-S' row) and scoring the progeny." Please leave proofs to the mathematicians.

"F6 mO/mC heterozygous females were brother-sister crossed against genotypic identical F6 males (Figure 2 and Figure 3, fourth row), resulting in F7 mO- and mC-only homozygotes that carry either only mO or only mC on both, the maternal and paternal chromosomes (Table 1, 'F6' row).” This paragraph is very redundant and the figure references are a bit mixed up. May I suggest: "F6 mO/mC heterozygous females were mated with genotypically identical F6 male siblings (Figure 2 and Figure 3, fourth row). The resulting F7 homozygotes carry either only mO or only mC on both maternal and paternal chromosomes (Table 1, 'F6' row, and Figure 2 and Figure 3, fifth row), in contrast to the F5 mO- and mC-only post recombination hemizygotes, which display the same phenotype but carry either mO or mC on the maternal chromosome (Figure 2 and Figure 3, third row).”

Mated with or crossed to is better than crossed against.

“Proven” vs. “demonstrated” again.

"Although some phenotype distributions differed from the theoretical Mendelian values" – is the observed deviation significant?

"functionality" should be “function”

"inversed genders should be “reversed” or “opposite”.

Discussion: “exemplarily shown” should be “exemplified”.

“desynchronized” should be “unsynchronized”.

“it will perform” should be “it can be performed”…

A list of points should be just first, second, third, not firstly, secondly, and remove the word "alike".

"This works independently of" should be "independent of".

“we used the AGOC vector concept to systematically creating functional homozygous *Tribolium* lines that are designed for fluorescence live imaging of embryonic development.” – should be: “We used the AGOC vector concept to systematically create functional homozygous Tribolium lines[…]”

"Automation devices, equipped with a fluorescence detection unit, can be used to sort embryos according to their genotype". Although their phenotype does represent their hemizygous genotype, they are still sorted directly by their “phenotype" please correct this.

Materials and methods: "this study utilizes a setup of vectors that are based on in silico designed and de novo synthetized sequences." Suggestion: “this study utilizes a set of vectors based on in silico design and de novo synthethesis.”

“(ii) the four-slots (#1to #4) cloning site…." Should be “(ii) the four-slot (#1 to #4) cloning site” and change “evolved” to “derived”.

Change “scarless” to “scarlessly”.

Change “evolution” to “development” or “construction”.

Change “synthetized” to “synthesized”.

"and for all ligations the T4 DNA ligase (M0202L, New England BioLabs or provided with the pGEM-T Easy vector).” Change this to

“…and T4 DNA ligase (….) for all ligations.”

“One transformation vector was created that allowed the expression of a mEmerald-labeled sianyltransferase ubiquitously throughout the whole embryonic development.” This sentence (and all other similar sentences) is a little confusing. Suggestion: “We created one transformation vector that ubiquitously drives expression of mEmerald-labeled sianyltransferase throughout embryonic development.”

Did you name Plain-White As Snow? If so, you should take credit, if not please reference.

“(10 object slides with about 50 embryos reach, totaling at approximately 500 embryos per round[…])” Do you mean 50 embryos each? Also what is an object slide? Is that the same as a cover slip?

“All crossings were single-pair with one female and one male adult". Suggestion: “All crosses were performed with single male female pairs.”

“The progeny was singularized during the larval or pupal stages”. Suggestion: “Individual progeny were placed in new growth vials and scored for the presence…”

The last paragraph of the subsection “Crossing procedure, insert number determination cross and homozygous viability cross” seems redundant with the results and figures. Since this is a methods paper that is ok and the mating scheme does not need to be reiterated in the Materials and methods section. Only a description of the mating procedure (vials/media) needs to be described here.

"This strain derives from the white-eyed Pearl mutant strain39, but also carries the light ocular diaphragm mutation40, so that the adult eyes appear completely non pigmented". Suggestion: “For this study we generated a double mutant strain containing the Pearl and LOC mutations that result in completely unpigmented or white eyes, which we named PWAS.”

Figure 3 legend:

"The percentage boxes show the experimental (and theoretical) ratios of the progeny that show the respective phenotype". Suggestion: “The percentage boxes indicate the experimental (and theoretical) ratios of the progeny that display the respective phenotype.”

Figure 4 legend:

“their appearance change”, change to “their change in appearance”.

"Detail images show the" change to “Higher magnification images show…”

Table 1. It is difficult to distinguish the Bold vs. normal fonts. May I suggest a light background fill for those cells?

Supplemental notes:

"to a high stress levels" should "be high stress levels".

Therefore, those transgenic individuals are principally homozygous viable, but a certain degree might fail to develop properly, resulting in biased homozygous viability cross ratios". Do you mean the line is essentially viable, but a certain percent of individuals might fail to develop properly, producing biased ratios in the progeny of the homozygous viability cross? If so please correct.

Figure 3—figure supplement 2 legend: Change "one Cre line" to "a specific Cre line".

Figure 1—figure supplement 4: Change “Evolution” to “Derivation”.

"Vectors belong either to one or to two types, as indicated by the differently colored backgrounds." In the very next sentence you mention five types. What are the two types?

Supplementary Figure 11 legend: Change “used” to “documented”.

Supplementary file 10: change “proved” to “provided”.

Supplementary file 1: in the left column first cell "one insertions" should be "one insertion"

(and in all relevant tables). In the right column percentage works for the theoretical rows but does not need to be repeated in the rest of the cells of that column. That column actually enumerates the total number or (n) in each row.

"Segregation of mO-mC of 60% or less was defined as the criteria for one insert." A little smoother and clearer: “Segregation of 60% or fewer mO-mC beetles[…]”

"this theoretical value corresponds to the case that the insertions occur on different chromosomes, which is more likely than two independent inserts on the same chromosome" change to:”…to the case in which the insertions” or”…to the case where the insertions”,

Supplementary file 4:

1) “The background line of choice, i.e. PWAS in our study. After a few weeks of adulthood, a replacement F0 is grown from a separate egg collection procedure to keep the line alive.” Suggestion: “The background line of choice, i.e. PWAS in our study. After a few weeks of adulthood, a replacement F0 was grown from a separate oviposition to keep the line alive.”

2) “Approximately 1-2% of surviving embryos become F1 mosaics. They typically do not show any fluorescence and can only be identified by crossing them against the wild-type and analyzing the progeny.” Suggestion: “Approximately 1-2% of surviving embryos become F1 mosaics, which typically do not display any fluorescence and can only be identified by crossing to wild-type and analyzing the progeny.”

3) “Until this generation, our route does not differ from the standard transgenic animal establishment procedure. Working cultures can be established from the F3 individuals that can be treated like classical transgenic lines and used in (preliminary) experiments. Independently, the AGOC procedure is initiated by crossing F3 mO-mC pre-recombination hemizygous female against the mCe helper homozygote male which express the Cre recombinase.” Suggestion: “Through this generation, our scheme does not differ from the standard procedures to generate transgenic animals. Working cultures can be established from the F3 individuals that can be treated like classical transgenic lines and used in (preliminary) experiments. Alternatively, the AGOC procedure can be initiated by crossing an F3 mO-mC pre-recombination hemizygous female to an mCe helper homozygous male that expresses the Cre recombinase.”

4) "The F4 mCe×mO-mC double hemizygotes are hybrids. Within this generation, recombination occurs and one of both markers is excised. This process happens mainly during later stages of development, so that the individuals typically show a patchy transformation marker expression pattern, which is apparent in the adult eyes" Suggestion: “The F4 mC; mO-mC double hemizygotes are hybrids. Within this generation, recombination occurs and one of the two markers in cis is excised. This process usually happens during later stages of development, such that individuals typically display patchy transformation marker expression in adult eyes.”

5) “In the F5 generation, the Cre-expressing transgene as well as one of the transformation markers segregate away from the other transformation marker. As the transformation marker is typically excised after most of the germline cells have differentiated, F5 mO- and mC-only post-recombination hemizygotes are obtained from a single cross.” May I suggest: “In the F5 generation, the Cre-expressing transgene as well as one transformation marker are removed. F5 mO- and mC-only post-recombination hemizygotes are obtained from a single cross since the transformation marker is typically excised after most of the germline cells have differentiated.”

The second sentence describes what occurs to generate F5 progeny, so it actually happens in the F4, if you do not wish to move the sentence, it will be clearer to reverse the two phrases, so that it is clear when/where the excision event occurs.

Supplementary figure legends non-annotated should be unlabeled.

Supplementary file 12.

What is mounting agarose it is only mentioned in this table?

What is recipe for agarose platform?

*Reviewer #3:*

This work represents a nice implementation of site-specific recombination combined with fluorescent marker genes to initially allow identification of hemizygous transformants, and subsequently (after Cre-mediated recombination) allow recognition of homozygous or hemizygous individuals. The work is thorough and rigorous. The presentation of data is more than adequate to justify the conclusions. The authors demonstrate the utility of their method in *Tribolium*, but it should be applicable to almost any organism.

However, the concept presented by these authors is not exceptionally novel. For instance, "brainbow" methods involve much the same sort of recombination to generate mutually exclusive marker expression, although it is not usually applied to the germline.

Other significant concerns relate to the Discussion, both at the beginning, and at the end.

1) It seems to me that the best justification for this work is the ability to easily recognize homozygous mutants. However, the authors do not discuss this until the very end of the paper. The authors should mention this aspect at the beginning, even in the Abstract. Although there are other reasons to produce homozygotes (stock keeping, higher yield in experimental crosses, etc.), identifying homozygous mutants certainly has to be at the top of the list. The authors don't actually generate mutants in this paper, which would have been a nice addition, but it isn't critical to justify the utility of the technique.

2) In the Discussion, I don't fully accept all the justifications for the technique. For instance, their first reason, that it saves manpower because genotyping can be done by simple examination under a fluorescence microscope, seems a bit dubious. It is easy – yes, but look at how many generations of crossing it took to get to that point – seven. If homozygotes can be produced in four generations by standard techniques, how much work did it actually save?

3) In many cases, it may not be necessary to create homozygous lines. If transgenic animals are recognized by fluorescence, and all that is needed is an animal with the transgene, then hemizygotes may be good enough to carry out planned experiments. In fact, in some cases it may be better to use the hemizygous individual. For instance, random transgene insertion could generate a recessive mutation that, unknowingly, affects the phenotype under examination.

4) Can homozygotes be identified by fluorescence intensity at generation 4, instead of adding three more generations to cross to Cre-expressing animals, generate reduction events, and then identify homozygotes by intercrossing?

5) I really have no idea what is meant by "synchronized genotyping" or "desynchronized genotyping".

---

## [Author Response]

Reviewer #1:In this paper, Strobl et al. present a germline transformation protocol using a novel vector they've developed, that streamlines the process of making a transgene insert into the genome homozygous. The concept is simple yet novel and applicable, with modifications, to a wide range of animal genetic models that lack other means of easily homozygosing transgenes, such as by the use of balancers or targeted insertion sites. The authors demonstrate their technique using vectors they've constructed and fluorescent eye markers in the flour beetle, Tribolium. The technique works, and would appear to be a useful one for this beetle and other model genetic systems that lack advanced genetic tools. Overall the paper is well put together; it's complete and easy to understand and includes the tests needed to validate the method presented. However the paper's methods and figures sections are much longer and more detailed than they need to be, so I suggest simplifying these to make the whole more accessible to the interested reader. Specific comments follow below:

We are grateful for the positive comments. We agree with reviewer #1 and shortened the Materials and methods section (from ~3,300 to ~2.500 words) and reduced the number of supplementary display items (from 28 to 20).

Please see also comment 7.

1) Subsection “Systematic creation of homozygous lines”, first paragraph. To put things in perspective, this would take 3 generations in Drosophila using balancers, so a 4 generation procedure in Tribolium is not revolutionary, but rather an incremental advance.

We removed the term ‘only’ and added a comparison with the balancer-based approach to the Discussion.

For a detailed rationale, see reviewer #3 comment 2.

2) The complete DNA sequences of the vectors should be made available as supplemental data for this paper. This is quite important to make the method accessible to the scientific community.

We attached the complete and comprehensively annotated sequences for all 25 vectors as both, GeneBank (.gb) and Geneious (.geneious) files as Supplementary file 3 compressed into a single zipped folder (.zip). A reference is in the Supplementary Material section.

3) Would it not make sense to invert one of the marker genes in the vector to prevent read-through? Please comment.

We decided to have both markers on the same strand in tail-to-head orientation simply because this results in ‘real’, heterozygous individuals in the F6 generation, i.e. mO and mC are in the same orientation. Thereby, we avoid a read through between the marker and the mEmerald expression cassette of the functional transgenic lines in the F7 homozygotes *after* recombination.

4) In Figure 2, why is the first box F3 rather than F1 or F2? Please comment. Why is there a dotted line connecting rows 3 and 4, rather than solid?

Our paper reports a proof-on-principle study, thus the workflow was designed to be as logical/comprehensible as possible. With the F2 × wild-type cross, we demonstrate that the respective lines carry only a single insertion *before* starting with the mating procedure. Theoretically, the number of insertions can be determined by quantitatively evaluating a F2 × helper line cross, but since this is the very first study on this topic, we prefer to present an easily understandable workflow.

Following the same rationale, we need to keep a non-recombined culture of each transgenic line so that colleagues can request those lines and reproduce our results. If the F2 female is directly mated with the homozygous helper line, all descendants undergo recombination and we cannot establish a non-recombined culture.

For colleagues who utilize our vector concept, both above mentioned limitations do not apply. They may mate F2 founders directly with the Cre-expressing helper line. They can determine the insert number from the F2 × helper line cross, and they do not necessarily need a non-recombined culture of their respective transgenic lines. However, if a non-recombined culture is desired, we recommend to use a male instead of a female individual as founder in the F2 generation. The male can be used to inseminate a wild-type female for 2-3 days and can then be mated with a Cre-expressing helper female to start the procedure. We started the very first run with a female founder in the F3 generation and decided – for consistency – to start the mating procedure always with a female individual.

Once our concept is broadly used for more insect species as well as adapted to model organisms, we intend to develop a protocol that outlines an optimized procedure in which the current F3 generation is ‘skipped’.

Please see also comment 4 and reviewer #3 comment 2.

The dotted line is now explained within the figure legend.

5) Figure 3 and Figure 4 are much more complex than necessary, and they didn't reproduce well. Please simplify these two figures.

We simplified Figure 3 by removing all images for the wild-type marker controls. The wild-type male in the F4, which is mated with the double hemizygous female, now also functions as the marker control (which is also indicated).

We think that the creation of homozygous transgenic lines for fluorescence live imaging assays is an important field of application for our concept, and that we also have to prove the functionality of the respective mEmerald-labeled Lifeact vectors beyond any doubt by providing a glimpse at our live imaging data. However, we understand that this data should not distract from the actual topic of our study, thus we strongly reduced the size of Figure 4. We are convinced that the figure is now at the practicable minimum – we still want to show that our vectors/transgenic lines perform adequately in long-term live imaging assays.

We are also convinced that the performance of novel transgenic lines in live imaging assays is of importance for the *Tribolium* community. In our study, we present the very first filamentous actin-labeled transgenic *Tribolium* lines and provide valuable information on two previously uncharacterized promoter sequences, Zen1 and ARP5.

6) The text in Table 1 is too small to read. Could this be presented in a more intuitive form?

We are well aware that Table 1 is relatively large, but it is the centerpiece of the study. We already designed the table as concisely as possible, and any restructuring would meddle with the distinctness. The final layout lies within the discretion of *eLife*. We will inquire if it is possible to print this table in landscape format, alternatively the small images that represent the genotypes can be removed.

7) The supplementary data are longer and more complex than necessary. Please include only what is necessary to convey the method to the reader.

We significantly reduced the number of supplementary display items. We simplified (similarly to Figure 3) and/or combined Supplementary Figure 3 and Figure 4 [now: Figure 3—figure supplement 2] and removed Supplementary Figures 6 and 11. We also simplified and combined Supplementary Tables 5 and 6, 1 and 9, 2 and 10, 3 and 11 as well as 13 and 16. In total, 8 supplementary display items were removed while the internal logic and overall consistency were improved.

8) In the Discussion, please include a list of the organisms in which this technique might be useful, and discuss how the vector would have to be modified to be used in these other organisms. Discussing these issues is important to ensure adoption of this method by the broader research community.

Reviewer #1 raises an important concern here. We already briefly discussed adaption to other model organisms, but we agree that our argumentation can be expanded. In the revised manuscript, we discuss adaption to other insect model organisms, zebrafish and mouse in more detail.

Reviewer #2:The authors describe important technical improvements to transgenic vector constructs that allow efficient production of homozygous transgenic strains. Their concept is applicable to any system in which one wishes to generate transgenic strains. The methodology is well written in the overall sense, but lacks the clarity that correct genetic terminology can bring. In the detailed comments I provide, I have suggested corrections for the first time I encountered an issue. However, it they should be applied throughout the narrative and legends.

We appreciate the very positive evaluation of our vector concept. Furthermore, we would like to express our gratitude for the comprehensive valuable suggestions on how to improve the nomenclature and scientific conciseness.

Impact statement:"allow to create homozygous transgenic lines systematically" is awkward, may I suggest:“allow us to systematically generate homozygous transgenic lines[…]”Or“allow the systematic construction of homozygous transgenic lines[…]”

We changed the impact statement accordingly.

Main text, first paragraph: “crossing setup[…]” Do you mean “crossing scheme”?

We changed the term ‘setup’ to ‘scheme’.

“allow to distinguish” is awkward, suggestion:“allow one to distinguish”.“purposely produced” is awkward, suggestion: “specifically designed”.

Both incongruities were corrected, the first sentence was rephrased to improve the overall conciseness, the second one was changed as suggested.

“phenotypically clearly distinguishable” – I understand the need for emphasis, but two adverbs seem like a little much.Clearly distinguishable is much smoother and just as informative. After all aren't all transformation markers producing a phenotype?

We agree with the argumentation of reviewer #2 here and removed ‘phenotypically’ from this sentence and from the Abstract.

"of both markers" a little clearer is "of the two markers"."The progeny was" – throughout the paper you are using progeny in the plural (you could substitute children, not child). Thus it should be progeny were.

We corrected respective occurrences throughout the whole manuscript.

"which contains a mOrange-based6 and a mCherry-based7 eye-specific8 transformation marker". Suggestion: “That contains both mOrange-based and mCherry-based eye-specific transformation markers".

The sentence was corrected accordingly.

“appropriate excitation and emission”. Suggestion: “appropriate excitation and emission filters”.

We rephrased the sentence as follows: ‘Both fluorescent proteins are spectrally separable by appropriate excitation bands and emission filters.’

"which were outcrossed against wild-type males" a little smoother and clearer "and these were crossed to".

We removed the term ‘outcrosses against’ completely throughout the manuscript and constantly use the term ‘mated with’ now.

Materials and methods:“study utilizes a setup of vectors that are based on in silico designed and de novo synthetized sequences”. Suggestion: “study utilizes a set of vectors based on in silico design and de novo synthesis.”

The sentence was shortened according to the suggestion.

“sublines were termed AGOC” – suggestion: “designated” or “called”.

We replaced ‘termed’ with ‘called’ in all occurrences except for the designation of vectors and genetic elements.

“Until this step, our route did not differ from most standard transgenic organisms establishment procedures.” A little smoother and clearer: “Through this step, our scheme did not differ from most standard procedures to generate transgenic organisms.”

We changed the sentence to ‘Up to this step, our scheme did not differ from most standard transgenic line establishment procedures.’

"which carry mO and mC consecutively on the[…]" change to "that carry mO and mC in tandem".

We changed the sentence to ‘…that carry mO and mC in cis configuration…’

"resulting in F4 mCe x mO-mC double hemizygotes in which Cre-mediated recombination occurs (Table 1, 'F3' row)" – this phrase actually follows from the previous sentence not the one to which it is attached. The genotype nomenclature is misleading, how about a completely new sentence:“In the resulting F4 double hemizygotes (Ce; mO-mC), Cre-mediated recombination occurs (Table 1, 'F3' row).” Also please note the cross occurs in the F3, the F4 should be described (Ce; mO-mC) not mCe x mO-mC). Please use a semicolon to denote non-linkage.

We rephrased this paragraph as well as the previous paragraph slightly. Information about the helper line is now provided in the previous paragraph.

Subsection “Systematic creation of homozygous lines”, same as above, the following is a cross not a genotype mCe x mO-mC. And "were outcrossed against wild-type males" Why are outcross and cross used interchangeably? It would be good to be strictly consistent in terminology usage.Also, inbreeding is not the same as selfing. (Beetles cannot be selfed, since they are not hermaphrodites.) The naïve reader will appreciate the clarity if you check through entire manuscript for term usage consistency.

We removed the term ‘outcross’ completely.

"F5 mO-only post-recombination hemizygous females were brother-sister crossed against F5 mC-only post recombination hemizygous males (Figure 2 and Figure 3, third row), resulting in F6 mO/mC heterozygotes, which carry once again both markers (Table 1, 'F5' row).” Since you are going to go into detail re cis/trans relationship of transgenes, why not use that terminology? May I suggest: “F5 mO-only post-recombination hemizygous females were mated to mC-only post recombination hemizygous male siblings (Figure 2 and Figure 3, third row), resulting in F6 mO/mC heterozygotes that carry both markers once again (Table 1, 'F5' row), but now in trans.”

We changed the sentence, but prefer to use the term ‘trans configuration’. We also appreciate the suggestion to use the term ‘siblings’ to ‘instead of the term ‘brother-sister’ to simplify this and similar sentences throughout our manuscript. We still prefer to keep the first sentence short and explain the marker configuration in a second sentence.

"In contrast to the F3 mO-mC pre-recombination hemizygotes, which show the same phenotype but carry both markers consecutively on the maternal chromosome, the F6 mO/mC heterozygotes carry mO on the maternal and mC on the paternal chromosome. This was proven by crossing F6 mO/mC heterozygous females against wild-type males (Table 1, 'F6-S' row)." Suggestion: “In contrast to the F3 mO-mC pre-recombination hemizygotes, which display the same phenotype but carry both markers in tandem on the maternal chromosome, the F6 mO/mC heterozygotes carry mO on the maternal and mC on the paternal chromosome. This was demonstrated by crossing F6 mO/mC heterozygous females against wild-type males (Table 1, 'F6-S' row) and scoring the progeny." Please leave proofs to the mathematicians.

We removed the first sentence (‘In contrast to[…]’) from the manuscript to avoid confusion. We changed the second sentence accordingly and used the terms ‘display’, ‘demonstrate’ and ‘scoring’ in other sections of our manuscript.

"F6 mO/mC heterozygous females were brother-sister crossed against genotypic identical F6 males (Figure 2 and Figure 3, fourth row), resulting in F7 mO- and mC-only homozygotes that carry either only mO or only mC on both, the maternal and paternal chromosomes (Table 1, 'F6' row).” This paragraph is very redundant and the figure references are a bit mixed up. May I suggest: "F6 mO/mC heterozygous females were mated with genotypically identical F6 male siblings (Figure 2 and Figure 3, fourth row). The resulting F7 homozygotes carry either only mO or only mC on both maternal and paternal chromosomes (Table 1, 'F6' row, and Figure 2 and Figure 3, fifth row), in contrast to the F5 mO- and mC-only post recombination hemizygotes, which display the same phenotype but carry either mO or mC on the maternal chromosome (Figure 2 and Figure 3, third row).”Mated with or crossed to is better than crossed against.

We simplified the paragraph by removing the ‘In contrast to[…]’ sentence to avoid confusion and changed the remaining sentences according to the suggestions.

For consistency, we stick to the term ‘mated with’.

“Proven” vs. “demonstrated” again.

We removed the term ‘prove’ nearly completely from the manuscript. We only kept the term ‘proof-of-principle’.

"Although some phenotype distributions differed from the theoretical Mendelian values" – is the observed deviation significant?

The differences are not significant. We rephrased the sentence to make our point clearer: “Throughout all generations, the subline-specific scores matched the expectations and no significant differences between the respective arithmetic means and the theoretical Mendelian ratios were found. Importantly, all expected phenotypes, and thus all expected genotypes, were found in all generations and F7 (mO/mO) as well as (mC/mC) homozygotes were obtained for all six AGOC sublines.”

We also added a sentence to the Materials and methods section that explains the statistical test that we performed, a notification to the description of Table 1 and the r respective supplementary tables. For convenience, the standard deviations were also added to the arithmetic mean rows in Table 1 and the respective supplementary tables.

"functionality" should be “function”

We corrected the issue

"inversed genders should be “reversed” or “opposite”.Discussion: “exemplarily shown” should be “exemplified”.“desynchronized” should be “unsynchronized”.“it will perform” should be “it can be performed”.A list of points should be just first, second, third, not firstly, secondly, and remove the word "alike"."This works independently of" should be "independent of".

All of those minor issues were resolved. We agree with reviewer #2 on the first concern, but changed ‘inversed genders’ to ‘swapped genders’ since we would like to avoid using genetic-related (e.g. ‘inverted terminal repeats’, ‘reverse transcription’) nomenclature here.

We also replaced Firstly etc. by Roman numerals within brackets.

Please see also reviewer #3 comment 5.

“we used the AGOC vector concept to systematically creating functional homozygous Tribolium lines that are designed for fluorescence live imaging of embryonic development.” – should be: “We used the AGOC vector concept to systematically create functional homozygous Tribolium lines[…]”

The mistake was corrected.

"Automation devices, equipped with a fluorescence detection unit, can be used to sort embryos according to their genotype". Although their phenotype does represent their hemizygous genotype, they are still sorted directly by their “phenotype" please correct this.

We are thankful for this important remark and changed the sentence to: ‘Automation devices, equipped with a fluorescence phenotype-adapted detection unit, in our case fluorescence, can be used to sort embryos organisms with different genotypes according to their markers’.

Materials and methods: "this study utilizes a setup of vectors that are based on in silico designed and de novo synthetized sequences." Suggestion: “this study utilizes a set of vectors based on in silico design and de novo synthethesis.”

We agree that the suggested shorter sentence also comes with an increase in legibility.

“(ii) the four-slots (#1to #4) cloning site[…]." Should be “(ii) the four-slot (#1 to #4) cloning site”.

We corrected the issue.

And change “evolved” to “derived”.Change “scarless” to “scarlessly”.Change “evolution” to “development” or “construction”.Change “synthetized” to “synthesized”.

All of those minor issues were corrected either by removing the respective sentences as part of the Materials and methods shortening procedure or by correcting the issue.

"and for all ligations the T4 DNA ligase (M0202L, New England BioLabs or provided with the pGEM-T Easy vector).” Change this to“[…]and T4 DNA ligase (….) for all ligations.”

We accept this suggestion and also changed the sentence accordingly.

“One transformation vector was created that allowed the expression of a mEmerald-labeled sianyltransferase ubiquitously throughout the whole embryonic development.” This sentence (and all other similar sentences) is a little confusing. Suggestion: “We created one transformation vector that ubiquitously drives expression of mEmerald-labeled sianyltransferase throughout embryonic development.”

We deleted this sentence (and other, similar sentences) as part of the Materials and methods shortening procedure.

Did you name Plain-White As Snow? If so, you should take credit, if not please reference.

We rephrased the sentence to make our point clearer.

“(10 object slides with about 50 embryos reach, totaling at approximately 500 embryos per round[…])” Do you mean 50 embryos each? Also what is an object slide? Is that the same as a cover slip?

We rephrased the sentence to make our point clearer.

“All crossings were single-pair with one female and one male adult". Suggestion: “All crosses were performed with single male female pairs.”

We adopted the suggestion.

“The progeny was singularized during the larval or pupal stages”. Suggestion: “Individual progeny were placed in new growth vials and scored for the presence[…]”

We also adopt this suggestion, but slightly rephrased the sentence.

The last paragraph of the subsection “Crossing procedure, insert number determination cross and homozygous viability cross” seems redundant with the results and figures. Since this is a methods paper that is ok and the mating scheme does not need to be reiterated in the Materials and methods section. Only a description of the mating procedure (vials/media) needs to be described here.

We followed the recommendation of reviewer #2 and replaced several sentences with a brief statement that the information is found within the Results section.

"This strain derives from the white-eyed Pearl mutant strain39, but also carries the light ocular diaphragm mutation40, so that the adult eyes appear completely non pigmented". Suggestion: “For this study we generated a double mutant strain containing the Pearl and LOC mutations that result in completely unpigmented or white eyes, which we named PWAS.”

We approve the rephrasing suggestion, but prefer to not abbreviate light ocular diaphragm.

Figure 3 legend:"The percentage boxes show the experimental (and theoretical) ratios of the progeny that show the respective phenotype". Suggestion: “The percentage boxes indicate the experimental (and theoretical) ratios of the progeny that display the respective phenotype.”

We accept the suggestion and adapt it for similar occasions.

Figure 4 legend:“their appearance change”, change to “their change in appearance”."Detail images show the" change to “Higher magnification images show[…]”

We changed ‘detail’ to ‘enlarged’. We prefer not to use the term ‘magnification’ here, since this might be misunderstood as ‘we switched to a higher magnification objective’.

Table 1. It is difficult to distinguish the Bold vs. normal fonts. May I suggest a light background fill for those cells?

We agree with reviewer #2 here, but we do not know of this will be in compliance with the *eLife* house style, since most of the tables that we saw within *eLife* articles only have shading in the header in the online version and no shading at all in the PDF version. If it is not possible, we would prefer to stick to bold font.

Supplemental notes:"to a high stress levels" should "be high stress levels".

We corrected the typo.

Therefore, those transgenic individuals are principally homozygous viable, but a certain degree might fail to develop properly, resulting in biased homozygous viability cross ratios". Do you mean the line is essentially viable, but a certain percent of individuals might fail to develop properly, producing biased ratios in the progeny of the homozygous viability cross? If so please correct.

Yes. We rephrased the sentence.

Figure 3—figure supplement 2 legend: Change "one Cre line" to "a specific Cre line".Figure 1—figure supplement 4: Change “Evolution” to “Derivation”.

Both corrections were made.

"Vectors belong either to one or to two types, as indicated by the differently colored backgrounds." In the very next sentence you mention five types. What are the two types?

We changed the sentences slightly to clarify that ‘individual vectors belong to either one or two of five different types’.

Supplementary Figure 11 legend: Change “used” to “documented”.Supplementary file 10: change “proved” to “provided”.

Supplementary Figure 11 was removed, so a change is no longer necessary.

The second remark was corrected.

Supplementary file 1: in the left column first cell "one insertions" should be "one insertion" (and in all relevant tables). In the right column percentage works for the theoretical rows but does not need to be repeated in the rest of the cells of that column. That column actually enumerates the total number or (n) in each row.

We corrected the typo and removed the ‘100%’ entries from the table and all other tables.

"Segregation of mO-mC of 60% or less was defined as the criteria for one insert." A little smoother and clearer: “Segregation of 60% or fewer mO-mC beetles[…]”

We changed the wording in this and the following supplementary table.

"this theoretical value corresponds to the case that the insertions occur on different chromosomes, which is more likely than two independent inserts on the same chromosome" change to: “[…]to the case in which the insertions” or”[…]to the case where the insertions”.

We changed the sentence accordingly.

Supplementary file 4:1) “The background line of choice, i.e. PWAS in our study. After a few weeks of adulthood, a replacement F0 is grown from a separate egg collection procedure to keep the line alive.” Suggestion: “The background line of choice, i.e. PWAS in our study. After a few weeks of adulthood, a replacement F0 was grown from a separate oviposition to keep the line alive.”

We accept the suggestion.

2) “Approximately 1-2% of surviving embryos become F1 mosaics. They typically do not show any fluorescence and can only be identified by crossing them against the wild-type and analyzing the progeny.” Suggestion: “Approximately 1-2% of surviving embryos become F1 mosaics, which typically do not display any fluorescence and can only be identified by crossing to wild-type and analyzing the progeny.”

We rephrased the sentence according to the suggestion and improved the overall description.

3) “Until this generation, our route does not differ from the standard transgenic animal establishment procedure. Working cultures can be established from the F3 individuals that can be treated like classical transgenic lines and used in (preliminary) experiments. Independently, the AGOC procedure is initiated by crossing F3 mO-mC pre-recombination hemizygous female against the mCe helper homozygote male which express the Cre recombinase.” Suggestion: “Through this generation, our scheme does not differ from the standard procedures to generate transgenic animals. Working cultures can be established from the F3 individuals that can be treated like classical transgenic lines and used in (preliminary) experiments. Alternatively, the AGOC procedure can be initiated by crossing an F3 mO-mC pre-recombination hemizygous female to an mCe helper homozygous male that expresses the Cre recombinase.”

We accept the suggestion, but modified the sentence slightly.

4) "The F4 mCe×mO-mC double hemizygotes are hybrids. Within this generation, recombination occurs and one of both markers is excised. This process happens mainly during later stages of development, so that the individuals typically show a patchy transformation marker expression pattern, which is apparent in the adult eyes" Suggestion: “The F4 mC; mO-mC double hemizygotes are hybrids. Within this generation, recombination occurs and one of the two markers in cis is excised. This process usually happens during later stages of development, such that individuals typically display patchy transformation marker expression in adult eyes.”

We approve the suggestion, rephrased the sentence, but stick to the term ‘cis configuration’.

5) “In the F5 generation, the Cre-expressing transgene as well as one of the transformation markers segregate away from the other transformation marker. As the transformation marker is typically excised after most of the germline cells have differentiated, F5 mO- and mC-only post-recombination hemizygotes are obtained from a single cross.” May I suggest: “In the F5 generation, the Cre-expressing transgene as well as one transformation marker are removed. F5 mO- and mC-only post-recombination hemizygotes are obtained from a single cross since the transformation marker is typically excised after most of the germline cells have differentiated.”The second sentence describes what occurs to generate F5 progeny, so it actually happens in the F4, if you do not wish to move the sentence, it will be clearer to reverse the two phrases, so that it is clear when/where the excision event occurs.Supplementary figure legends non-annotated should be unlabeled.

We exchanged the terms as suggested.

Supplementary file 12.What is mounting agarose it is only mentioned in this table?

All developmental biology-associated live imaging assays with light sheet-based fluorescence microscopy use mounting agarose to form a stable mounting matrix (a column, a block, a cup, a hemisphere, a thin film, a pocket…). Protocols for insect embryos are actually well-established (*Drosophila*: Keller et al. 2011, Cold Spring Harbor Protocols; Schmied and Tomancak 2016, Methods in Molecular Biology), and two protocols for *Tribolium* are cited within the Materials and methods section. We improved the table by adding another row that references the detailed mounting technique.

What is recipe for agarose platform?

We made slight adjustments to the sentence to clarify our point.

Reviewer #3:This work represents a nice implementation of site-specific recombination combined with fluorescent marker genes to initially allow identification of hemizygous transformants, and subsequently (after Cre-mediated recombination) allow recognition of homozygous or hemizygous individuals. The work is thorough and rigorous. The presentation of data is more than adequate to justify the conclusions. The authors demonstrate the utility of their method in Tribolium, but it should be applicable to almost any organism.However, the concept presented by these authors is not exceptionally novel. For instance, "brainbow" methods involve much the same sort of recombination to generate mutually exclusive marker expression, although it is not usually applied to the germline.

We only partially agree with reviewer #3 here. The functions of both concepts are entirely different and not comparable. Brainbow was developed to aid on the ‘cellular level’. It has nothing to do with genotyping, husbandry and the systematic creation of homozygous transgenic lines but supports the identification and characterization process of individual neurons and their respective dendrites and axons in a single organism. Our concept was developed to aid on the ‘organismal level’. It has nothing to do with differences amongst individual cells but facilitates the handling of the many multi-cellular individuals that are required to maintain the line and obtain descendants with the desired genotype.

Other significant concerns relate to the Discussion, both at the beginning, and at the end.1) It seems to me that the best justification for this work is the ability to easily recognize homozygous mutants. However, the authors do not discuss this until the very end of the paper. The authors should mention this aspect at the beginning, even in the Abstract. Although there are other reasons to produce homozygotes (stock keeping, higher yield in experimental crosses, etc.), identifying homozygous mutants certainly has to be at the top of the list. The authors don't actually generate mutants in this paper, which would have been a nice addition, but it isn't critical to justify the utility of the technique.

We strongly agree with reviewer #3 that workflows, which involve the creation of homozygous mutants, independent the actual method – e.g. large scale insertional mutagenesis (Trauner et al. 2009, BMC Biology) or genome engineering (Gilles et al., 2015) – can benefit from our vector concept, which does not only apply to *Tribolium* but also to other model organisms. A study in which the AGOC vector concept is used as part of a mutagenesis / knock-out assay can be expected in the near future. We added a short sentence to the Abstract.

2) In the Discussion, I don't fully accept all the justifications for the technique. For instance, their first reason, that it saves manpower because genotyping can be done by simple examination under a fluorescence microscope, seems a bit dubious. It is easy – yes, but look at how many generations of crossing it took to get to that point – seven. If homozygotes can be produced in four generations by standard techniques, how much work did it actually save?

We agree with reviewer #3 that *in certain scenarios, for some aspects,* our concept is rivaled by other techniques, ‘waiting time’ being the obvious one. We are well aware of this, thus we intentionally avoid the term ‘time’ and rather justify our approach by discussing manpower (i.e. the total ‘working time’ the scientist is busy performing the respective assays) and consumables. Since genotyping and homozygosing is an important, yet very complex and extremely heterogeneous topic (which model organism, which transgenesis approach and which genotyping / homozygosing strategy is used?), a comprehensive comparison lies beyond the scope of the discussion. Our concept complements and improves the arsenal of genotyping and homozygosing techniques. We put a lot of effort into explaining the architecture and functionality as detailed as possible so that interested readers can decide whether our suggestion benefits their workflow or whether another approach should be chosen.

Since *eLife* also publishes the decision letter and the author response, we would like to provide a few more thoughts that may be of relevance for interested colleagues:

- In *Drosophila*, when site-specific integration is used, balancer-based homozygosing requires less ‘waiting time’ in comparison to our concept. However, in *Tribolium* and mouse, only very few, and for zebrafish, basically no balancers are available. Additionally, usage of balancers increases recombination in non-balancer regions, the respective balancer lines have to be maintained in the laboratory (or obtained from a stock center or colleague) and certain balancers may also not be available for the background strain of choice. Our concept was established with universality in mind and thus has to be judged from multiple points of view and by considering all relevant circumstances.

- When random integration is performed, the few available balancers are only a convenient choice when the insertion site is known. Genotyping via genetic assays also requires data on the insertion site, leaving test crossing assays as the only option. Test crossing by itself also has severe drawbacks.

- When a large number of newly established transgenic lines have to be handled, ‘waiting time’ is usually only a minor issue, while manpower as well as resources become major challenges and thus also the scope-limiting factors. Recently, we also established a genetic assay-based non-lethal genotyping protocol for *Tribolium* (Strobl et al., 2017) and are thus able to compare both approaches at first hand, confirming that the savings in manpower and resources are significant.

- This is a proof-of-principle study, therefore we decided to demonstrate the functionality of our concept as ‘clean’ as possible for 7 generations. At least three ‘quick and dirty shortcuts’ can be considered:

i) The F3 generation is optional, F2 founders can be directly mated with the Cre-expressing helper lines. Since we have to keep a non-recombination culture of all transgenic lines for reasons of reproduction, we had to add another generation.

ii) F4 [now F3] (mCe; mO-mC) double hemizygotes can be mated with genotypically identical siblings, resulting in 6.25% F6 [now F4] (mO/mC) heterozygotes.

iii) F6 [now F4] (mO/mC) heterozygotes are, in basically all experimental scenarios, functionally identical with F7 (mO/mO) homozygotes / F7 (mC/mC) homozygotes.

- Workflow complexity should also not be underestimated. After a brief introduction, the procedure that we outline within our manuscript can be robustly conducted by student assistants.

See also reviewer #1 comments 1 and 4.

We removed one occurrence of the term ‘time’ and rephrased the respective sentence. We also mention the balancer-based approach briefly within the Discussion.

3) In many cases, it may not be necessary to create homozygous lines. If transgenic animals are recognized by fluorescence, and all that is needed is an animal with the transgene, then hemizygotes may be good enough to carry out planned experiments. In fact, in some cases it may be better to use the hemizygous individual. For instance, random transgene insertion could generate a recessive mutation that, unknowingly, affects the phenotype under examination.

We only partially agree with reviewer #3 here. In theory, there are assays in which hemizygotes are the preferred choice and other assays in which homozygotes are the preferred choice. However, e.g. in fluorescence live imaging assays using transgenic lines, there are typically two practical cases: either the embryo collection culture is homozygous, and thus all progeny is homozygous, or the culture is mixed (hemi- and homozygotes, eventually also wild-types), and the genotype of the descendants may remain unknown. This is absolutely not recommended since it adds another uncontrollable parameter to the assay which might have a substantial influence on the results.

The concern that reviewer #3 raises here is actually one of the major advantages of our concept. The experimenter can systematically evaluate if the transgene insertion into a random location leads to a phenotype when homozygously present. Thus, the effect becomes known, and the experimenter can continue to work with hemizygous individuals or create more transgenic sublines in which the transgene does not provoke a phenotype when homozygously present.

4) Can homozygotes be identified by fluorescence intensity at generation 4, instead of adding three more generations to cross to Cre-expressing animals, generate reduction events, and then identify homozygotes by intercrossing?

In *Tribolium* (and probably also all in other model organisms) this would not reliably work. For most of the transgenic lines, the mean expression levels of the markers do not obviously differ between hemizygous and homozygous individuals. For a few other lines, the *mean* expression levels of homozygotes appear to be slightly higher than for hemizygotes. However, due to individual plasticity and most probably other factors such as age and maybe also gender, homozygotes and hemizygotes have broad signal strength windows with a large overlap. Thus, some (weakly expressing) homozygotes show less signal than some (strongly expressing) hemizygotes. Also, such an approach would not work very well in conjunction with automation.

In biology, it is always problematic to reliably distinguish between ‘some’ and ‘some more’, especially when both windows have a large overlap. With our concept, the experimenter has to differentiate between ‘none’ and ‘some’, which do not overlap at all. This is the most convenient and most reliable approach.

5) I really have no idea what is meant by "synchronized genotyping" or "desynchronized genotyping".

We changed the term ‘desynchronized’ to ‘unsynchronized’ and improved the sentences by clarifying that we refer to multiple individuals.